Brief Communication

# Mechanism of mammalian transcriptional repression by noncoding RNA

Katarína Tlučková[1,2], Beata Kaczmarek[1,2], Anita Salmazo[1] & Carrie Bernecky [1] ✉

Transcription by RNA polymerase II (Pol II) can be repressed by noncoding RNA, including the human RNA Alu. However, the mechanism by which endogenous RNAs repress transcription remains unclear. Here we present cryogenic-electron microscopy structures of Pol II bound to Alu RNA, which reveal that Alu RNA mimics how DNA and RNA bind to Pol II during transcription elongation. Further, we show how distinct domains of the general transcription factor TFIIF control repressive activity. Together, we reveal how a noncoding RNA can regulate mammalian gene expression.

RNA polymerase II (Pol II) is the 12-subunit eukaryotic enzyme that generates messenger RNA and is thus a focal point for the regulation of transcription. The activity of Pol II is tightly regulated throughout the transcription cycle by association with various accessory factors, including single proteins, multi-subunit protein complexes and RNAs[1]. Whereas protein regulators of transcription have been extensively studied, RNA-based mechanisms for regulation remain poorly understood.

Human Alu RNA has been identified as a natural inhibitor of transcription, blocking transcription of specific genes during heat shock[2]. Alu RNA is transcribed by Pol III as single units from short interspersed elements, abundant genomic repeats. These free Alu RNAs are present at low levels under normal conditions, but levels increase in response to cellular stress[3]. Elevated levels of Alu RNA are able to repress Pol II transcription and, after heat shock, Alu RNA can be found at the promoters of repressed genes, consistent with the direct binding to Pol II observed in vitro[2]. Biochemical analysis revealed that Alu RNA binds directly to two molecules Pol II via independent interactions with the two halves of Alu RNA, Alu-left arm (Alu-LA, also known as scAlu) and Alu-right arm (Alu-RA)[2] (Fig. 1a). Although both Alu-LA and Alu-RA bind to Pol II with high affinity, only Alu-RA is able to inhibit transcription in the presence of the heterodimeric general transcription factor TFIIF, comprising the proteins RAP74 and RAP30. It has been shown by low-resolution (~25 Å) cryogenic-electron microscopy (cryo-EM) reconstructions[4] that both Alu-LA and Alu-RA localize to the Pol II DNA-binding cleft. Because the RNA secondary structure could not be resolved, it remains unknown how Alu RNAs are able to form high-affinity contacts with Pol II, thus allowing them to repress within an endogenous context. Additionally, the mechanism by which the repression of only one of these very similar Alu RNA halves is affected by TFIIF remains to be elucidated.

## Results and discussion
### Structural analysis of Pol II bound to Alu RNA

To investigate the principles determining Pol II inhibitory activity of Alu RNA, we reconstituted complexes of mammalian Pol II bound to in vitro transcribed and refolded Alu-LA and Alu-RA (Extended Data Fig. 1). Our initial attempts at cryo-EM sample preparation using holey carbon or thin carbon-coated electron microscopy grids revealed that the samples displayed biased particle orientations, hindering high resolution analysis. Mild BS3 or glutaraldehyde crosslinking, which improved particle orientations for the Pol II elongation complex (EC)[5], resulted in dissociation of Alu RNA. Preparation of sample using graphene oxide-coated holey carbon grids[6] resulted in less biased particle orientations and allowed structure determination of Pol II–Alu-LA and Pol II–Alu-RA to nominal resolutions of 2.4 and 2.5 Å (Table 1). For the Pol II–Alu-LA complex, the RNA could be resolved to approximately 5–8 Å. Similarly, Alu-RA within the Pol II–Alu-RA complex was resolved to approximately 5–8 Å. In both complexes, the Pol II clamp and stalk domains could be well resolved with further classification (Extended Data Figs. 2 and 3).

The resolution of the RNA precluded building of an atomic model, but did reveal helical structure and the binding location of Alu RNA. Both Alu-LA and Alu-RA bind in the Pol II DNA-binding cleft, adopting a conformation similar to that of DNA and RNA within the Pol II EC (Fig. 1b,c). Comparison to the Pol II EC revealed helical Alu RNA density overlapping the positions of the DNA–RNA hybrid, specific to transcribing complexes, and downstream DNA. Additional density partially overlapped the positions of upstream DNA and exiting RNA within the Pol II EC. This can explain how Alu RNA would block productive engagement with promoter DNA during transcription initiation, but could still allow assembly of a preinitiation complex of altered topology through TFIIB- and TBP-promoter DNA contacts[7].

[1]Institute of Science and Technology Austria (ISTA), Klosterneuburg, Austria. [2]These authors contributed equally: Katarína Tlučková, Beata Kaczmarek.
✉e-mail: carrie.bernecky@ist.ac.at

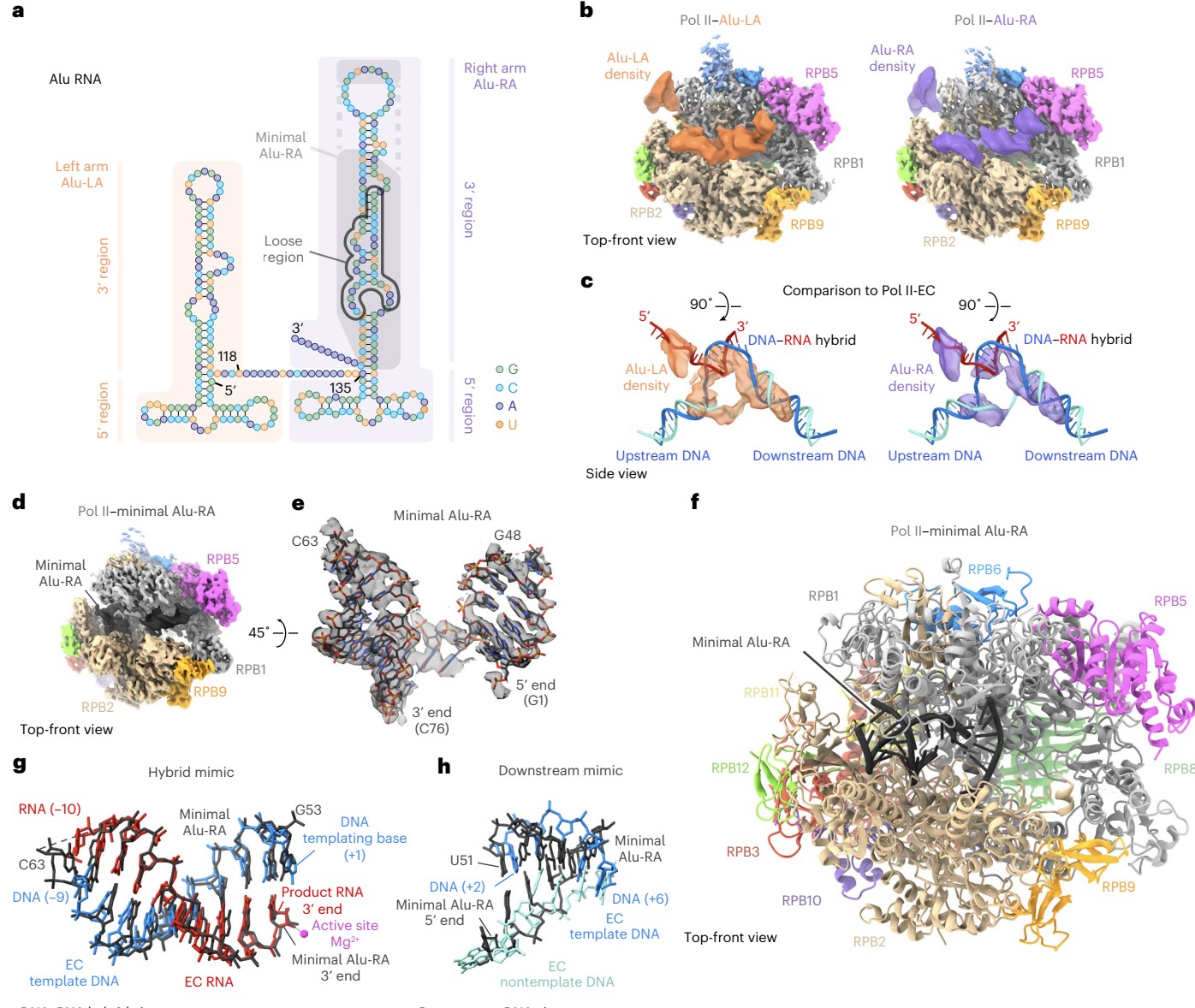

**Fig. 1 | Structural analysis of Pol II bound to Alu RNA. a**, Schematic showing the secondary structure and domain organization of Alu RNA. Alu-LA (orange shading), Alu-RA (purple shading), Alu-RA minimal region (gray shading) and Alu-RA loose region (gray outline) are indicated. The 5′ and 3′ regions of Alu-LA and Alu-RA are indicated with orange and purple bars. Guanosine, green; cytidine, cyan; adenosine, blue; uridine, orange. **b**, Top-front view of Pol II–Alu-LA and -Alu-RA reconstructions (not *B*-factor sharpened), with 8-Å-filtered Alu-LA (orange) and Alu-RA (purple) difference densities overlaid. Pol II is colored by subunit. **c**, Side view of 8-Å-filtered Alu-LA and Alu-RA densities compared to the path of nucleic acids in a transcription EC (PDB ID 5OIK)[27]. **d–h**, Structure of Pol II–minimal Alu-RA. **d**, Top-front view of the locally filtered Pol II–minimal Alu-RA alternative conformation density. **e**, A rotated view of the modeled region of minimal Alu-RA overlaid with the electron microscopy density (DeepEMhancer postprocessed for visualization). **f**, Top-front view of the Pol II–minimal Alu-RA structure. **g**, A view of the DNA–RNA hybrid-like region of minimal Alu-RA and the same region of a transcription EC (PDB ID 5OIK)[27]. **h**, As in **g**, but focused on the downstream DNA region.

Unlike in the EC, the Pol II clamp domain did not close over the nucleic acids in a stable conformation. Fork loop 2 within the DNA-binding cleft, which interacts with the melted nontemplate DNA strand in the EC, was rearranged from the conformation observed in the EC to that observed in Pol II lacking DNA or RNA[5], consistent with the absence of a melted nucleic acid strand in the Alu RNA structures (Extended Data Fig. 3g). Altogether, despite Alu RNAs adopting an EC-like conformation, the Pol II enzyme was observed in an inactive state, consistent with an enzyme–nucleic acid complex not competent for elongation.

If Alu RNA resolution was restricted by intrinsic flexibility, we reasoned that a shorter RNA construct could overcome that limitation.

To that end, we prepared a minimal version of Alu-RA (Fig. 1a) containing the structured elements previously shown to be sufficient for both Pol II binding and repression[2]. We solved the structure of the Pol II–minimal Alu-RA complex to a nominal resolution of 3.1 Å, and observed that the quality of the RNA density in the initial map was greatly improved. Three-dimensional (3D) classification revealed two minimal Alu-RA conformations (Extended Data Fig. 4). The first RNA conformation could be resolved to an intermediate resolution of approximately 5–8 Å, and corresponded to an Alu-RA–RNA-like conformation. This allowed the putative assignment of RNA regions to the Alu-RA density (Extended Data Fig. 4i). The second conformation could be resolved to a higher resolution of 3–6 Å, in which individual

**Table 1 | Cryo-EM data collection, refinement and validation statistics**

| | Pol II–Alu-LA (EMDB-18367) | Pol II–Alu-RA (EMDB-18375), (PDB 8QEP) | Pol II–minimal Alu-RA, canonical conformation (EMDB-18371) | Pol II–minimal Alu-RA, alternative conformation (EMDB-18376), (PDB 8QEQ) |
|---|---|---|---|---|
| **Data collection and processing** | | | | |
| Magnification | ×105,000 | ×105,000 | ×105,000 | ×105,000 |
| Voltage (kV) | 300 | 300 | 300 | 300 |
| Electron exposure (e⁻/Å²) | 40 | 40 | 41 | 41 |
| Exposure rate (e⁻/pix/s) | 13.8 | 12.3, 11.8, 15.0 | 16.4 | 16.4 |
| Defocus range (μm) | −0.2 to −2.0 | −0.2 to −2.2 | −0.1 to −2.2 | −0.1 to −2.2 |
| Pixel size (Å) | 0.835 | 0.835 | 0.835 | 0.835 |
| Symmetry imposed | C1 | C1 | C1 | C1 |
| Number of micrographs | 16,470 | 29,262 | 7,588 | 7,588 |
| Initial particle images (no.) | 1,402,832 | 2,114,883 | 704,869 | 704,869 |
| Final particle images (no.) | 152,647 | 857,265 | 72,783 | 90,778 |
| Map resolution (Å) | 2.4 | 2.5 | 3.2 | 3.1 |
| FSC threshold | 0.143 | 0.143 | 0.143 | 0.143 |
| Map resolution range (Å) | 2.3–8.0 | 2.3–8.0 | 2.7–8.0 | 2.7–8.0 |
| **Refinement** | | | | |
| Initial model used (PDB code) | | 5OIK | | 5OIK |
| Model resolution (Å) | | 2.6 | | 3.2 |
| FSC threshold | | 0.5 | | 0.5 |
| Map sharpening $B$ factor (Å²) | −40.63 | −50.93 | −46.12 | −39.56 |
| Model composition | | | | |
| Nonhydrogen atoms | | 25,496 | | 29,257 |
| Protein residues | | 3,177 | | 3,575 |
| Ligands | | 4 | | 8 |
| $B$ factors (Å²) | | | | |
| Protein | | 39.65 | | 49.14 |
| Ligand | | 70.50 | | 92.84 |
| R.m.s. deviations | | | | |
| Bond lengths (Å) | | 0.003 | | 0.004 |
| Bond angles (°) | | 0.725 | | 0.748 |
| **Validation** | | | | |
| MolProbity score | | 1.40 | | 1.44 |
| Clashscore | | 2.24 | | 2.40 |
| Poor rotamers (%) | | 1.63 | | 1.39 |
| Ramachandran plot | | | | |
| Favored (%) | | 96.40 | | 95.50 |
| Allowed (%) | | 3.54 | | 4.22 |
| Disallowed (%) | | 0.06 | | 0.28 |

bases could be distinguished (Fig. 1d,e). A model of Pol II and the well-resolved region of minimal Alu-RA in the alternative conformation was generated using a combination of de novo model building with DeepTracer[8] and secondary structure-based manual building[9] (Fig. 1f). The alternative conformation even more closely mimicked EC nucleic acids (Fig. 1g,h). Comparison of the Pol II–minimal Alu-RA structure to that of *Saccharomyces cerevisiae* Pol II bound to Fc aptamer[10], a synthetic RNA inhibitor, revealed that although both RNAs bind to a region overlapping that bound by the DNA–RNA hybrid, only minimal Alu-RA forms essentially the same interactions as those found within the EC (Extended Data Fig. 5a,b). Thus, stable binding of mammalian Pol II to structured RNA likely involves contacts similar to those formed within an EC.

Analogous to Alu RNA, bacterial 6S RNA is a natural noncoding RNA that directly binds the bacterial RNA polymerase and inhibits transcription from σ70-dependent housekeeping promoters. Both Alu and 6S RNA inhibit promoter DNA engagement. A previous near-atomic structure of *Escherichia coli* RNA polymerase in complex with 6S RNA showed that this RNA adopted a conformation mimicking that of open promoter DNA[11]. This contrasts with our observation that Alu RNA mimics EC nucleic acids (Fig. 1c). Previous results have shown the importance of a loosely structured region within Alu-RA for repressive activity[2] (Fig. 1a). We conclude that loosely base paired regions within Alu RNAs are important not as a mimic of open promoter DNA (Extended Data Fig. 5c), but speculate that it allows the RNA the flexibility to adopt a conformation matching that of the EC DNA–RNA

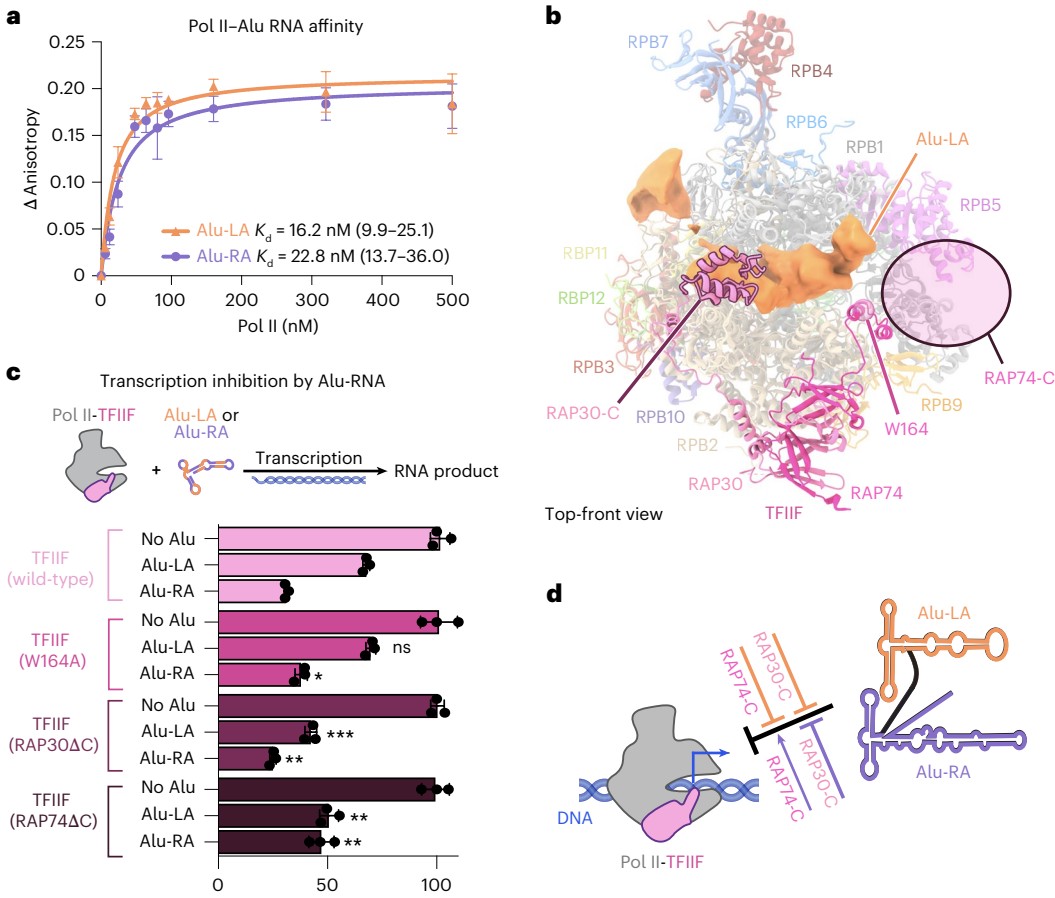

**Fig. 2 | Effects of TFIIF and its domains on activity of Pol II–Alu RNA complexes. a**, Fluorescence anisotropy experiments reveal Alu-LA and Alu-RA binding affinity to Pol II. Results represent analysis of three independently prepared reactions. Data are plotted as mean values ± standard deviation. **b**, Alu-LA density overlaid on Pol II-TFIIF as found within a transcription initiation open complex (PDB ID 5IYB)[25]. Pol II subunits are shown as semitransparent ribbons. The mutated regions of the TFIIF constructs used in **c** are indicated. **c**, Transcription assays carried out in the presence of the indicated TFIIF variant and either no RNA, Alu-LA or Alu-RA. Results represent analysis of three

independently prepared reactions. Data are presented as mean values ± standard deviation, with individual data points shown as black circles. Achieved significance level relative to the wild-type TFIIF condition was calculated using an unpaired, two-tailed *t*-test: ***$P \leq 0.001$; **$P \leq 0.01$; *$P \leq 0.05$; NS, $P > 0.05$. Exact *P* values are as follows: TFIIF (W164A), Alu-LA $P = 0.0169$, Alu-RA $P = 0.2798$; TFIIF (RAP30ΔC), Alu-LA $P = 0.0002$, Alu-RA $P = 0.0040$; TFIIF (RAP74ΔC), Alu-LA $P = 0.0030$, Alu-RA $P = 0.0089$. **d**, Schematic representation of the effects of TFIIF domains on the repressive activity of Alu-LA and Alu-RA.

hybrid and downstream DNA. This is consistent with previous observations that, unlike bacterial promoter complexes that can form a stable complex with just RNA polymerase and a sigma factor[12,13], eukaryotic initiation complexes require a large number of transcription factors to stabilize the interaction with DNA[14] and are dynamic[15]. Promoter complexes lacking accessory factors or a nascent RNA chain are unstable[16]. Thus, the more stable EC[16–18] would be an effective mammalian intermediate to mimic.

### TFIIF domains differentially affect transcriptional repression by Alu RNA

Although Alu-LA and Alu-RA form similar structures, previous studies have shown that Alu-LA is more labile in the presence of TFIIF[19]. To investigate these differences in Pol II–Alu RNA complex binding, we first measured binding affinities of Alu-LA and Alu-RA to Pol II using fluorescence anisotropy. We observed similarly high affinities for Alu-LA and Alu-RA when investigated in the presence of 150 mM monovalent salt (Fig. 2a). The similar affinities were consistent with previous reports in which the complex was investigated in low salt conditions[2], despite a decrease of the apparent binding strength by approximately tenfold.

Inspection of the Alu structures shows that TFIIF would not sterically clash with the resolved RNA elements (Fig. 2b)[20]. TFIIF has been

shown to inhibit nonspecific DNA binding by Pol II, with the small subunit of TFIIF implicated as particularly important[21]. It has also been suggested that a negatively charged region of the large subunit of TFIIF may be responsible for repelling nonspecific DNA from the active site cleft[22]. To investigate how these regions of TFIIF may differentially affect Alu-LA and Alu-RA, we performed transcription inhibition assays in the presence of TFIIF or TFIIF mutants (Fig. 2c and Extended Data Fig. 6a–e). As expected, we observed that in the presence of TFIIF, Alu-RA inhibits transcription more strongly than Alu-LA or an unrelated, unstructured RNA (Extended Data Fig. 6d)[23]. A TFIIF mutant within the charged region helix of the large subunit, TFIIF (W164A), known to be defective in transcription initiation but maintaining both domains thought to be important in the ability of TFIIF to inhibit nonspecific DNA binding[24], behaved similar to wild-type TFIIF. We next investigated the effect of the winged helix domain of the small TFIIF subunit, a domain that binds upstream promoter DNA within the transcription initiation complex[25] and is important for transcription initiation activity[26]. TFIIF (RAP30ΔC) was unable to relieve transcriptional repression by either Alu construct. Last, we tested a TFIIF mutant lacking the C terminus of the largest subunit, TFIIF (RAP74ΔC). The deleted region contained the negatively charged region and a winged helix domain. In the presence of TFIIF (RAP74ΔC), Alu-RA transcriptional repression was partially

relieved relative to wild type, suggesting that either the charged low complexity domain or the winged helix, or both together, can enhance Alu-RA repressive activity. In contrast, Alu-LA transcriptional repression activity increased slightly, suggesting that the C-terminal region is deleterious for the repressive activity of this Alu construct.

To test whether direct interactions between TFIIF and Alu RNA could be involved in transcriptional derepression, we investigated the ability of TFIIF to interact with Alu-LA and Alu-RA and found that wild-type TFIIF bound to both Alu halves. Furthermore, both TFIIF (RAP30ΔC) and TFIIF (RAP74ΔC) could interact with Alu-LA and Alu-RA, with TFIIF (RAP74ΔC) displaying reduced binding relative to wild-type TFIIF (Extended Data Fig. 6f,g). Based on previously reported initiation complex structures[25], the RAP30 winged helix domain would localize to the upstream DNA region, in close proximity to the putative location of the more conserved 5′ regions of Alu-LA and Alu-RA. Consistent with a localization near conserved RNA regions, we observed similar effects of TFIIF (RAP30ΔC) on Alu-LA and Alu-RA repression. We speculate that the shorter, RNA bulge-containing 3′ region of Alu-LA may sensitize the Pol II–Alu-LA complex to disruption by the RAP74 C terminus, whereas the longer 3′ region of Alu-RA could allow association of the RAP74 C terminus with RNA without Pol II–Alu-RA disruption. Altogether, these results indicate that the winged helix of the small TFIIF subunit RAP30 acts similarly on the repressive and nonrepressive RNAs, whereas the C terminus of the large subunit RAP74 contributes to the distinct activities of Alu-LA and Alu-RA.

In summary, we found that Alu RNA is able to tightly bind Pol II via mimicry of EC nucleic acids, in contrast to the promoter-like mimicry reported in the bacterial system[11]. Although both halves of Alu RNA adopt the same EC-like structure when bound to Pol II, transcriptional repressive activity of Alu RNA relies on the differential response to distinct TFIIF domains, potentially through TFIIF–RNA binding activity. We propose that EC mimicry and TFIIF domain-dependent activity may be a general mechanism by which structured noncoding RNAs regulate mammalian transcription.

## Online content

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

## Methods

### Cloning and constructs

The full-length Alu sequence from complementary DNA clone TS 103 (refs. [2],[28]) containing an $A_{10}$ poly(A) tail was purchased as a gBlock (IDT) and cloned into a modified pSP64 plasmid (Promega) containing an autocleavable 3′ terminal hepatitis delta virus ribozyme[29]. Alu-LA (nucleotides 1–118) was PCR amplified from the full-length construct and Alu-RA (nucleotides 134–291 containing a 5′ hammerhead ribozyme sequence[30]) was purchased as a gBlock (IDT). Both constructs were cloned into the modified pSP64 plasmids. The minimal Alu-RA sequence (nucleotides 183–216, 230–238 and 249–281) was ordered as a duplexed DNA containing a 5′ T7 promoter (IDT). A plasmid encoding wild-type $His_{10}$-$Arg_7$-3C-SUMO-TFIIF was a gift from D. Slade[31]. TFIIF mutants were created by round-the-horn site-directed mutagenesis. The TFIIF mutant W164A contains a single point mutation, W164A, in the charged region of GTF2F1 (RAP74)[32]. In TFIIF (RAP74ΔC), amino acids 181–517, encompassing the unstructured region and winged helix of GTF2F1 (RAP74), were deleted. In TFIIF (RAP30ΔC), amino acids 175–249, encompassing the GTF2F2 winged helix, were deleted, with a short, vector-derived short C-terminal extension remaining ('LSVF-FLFPFSPRSPAFPVKL'). The sequences of all oligonucleotide primers used in this study can be found in the Supplementary Information.

### RNA production

Plasmids were used to generate a PCR product containing Alu-LA or Alu-RA sequences and ribozyme as detailed above, which was then used as a template for in vitro transcription using T7 RNAP (prepared in-house[33]). The templates for an unstructured 151-nucleotide-long RNA (RNA 151) (sequence 5′- GAACGACAUCAUAA CAUUUGAACAAGAAUACCAAUACAUAAAUCCAUUAAGAACGACA UCAUA ACAUUUGA ACAACA AUA AACAUACAUAA AUCCAUC A AGA ACG AGA UCA UAA CAUUUG AACA AGA AUAUAUA UACAUAAAUCCAUCAUA-3′) and minimal Alu-RA were produced by PCR, and RNA was produced via runoff transcription. The PCR template was removed by RQ1 RNase-free DNase I treatment (Promega). RNAs were gel purified and ethanol precipitated. Hepatitis delta virus ribozyme cleavage resulted in 3′ cyclic phosphates, which were removed by T4 PNK (NEB) treatment (100 mM MES-NaOH pH 5.5 at room temperature, 10 mM $MgCl_2$, 10 mM 2-mercaptoethanol, 300 mM NaCl, 1 U μl$^{-1}$ RiboLock RNase Inhibitor (Thermo Scientific)) as previously described[34]. RNA was again gel purified and ethanol precipitated. RNA 151 was subjected to an additional round of DNase I treatment and repurified using a Monarch RNA clean-up kit (NEB).

### RNA labeling

RNAs were 3′ end-labeled by ligation of pCp-ATTO-488. RNA (100 pmol in water) was heated for 3 min at 95 °C, cooled at 25 °C for 10 min and labeled in 50 μl of reaction containing a fivefold molar excess of pCp-ATTO-488 (Jena Bioscience), 1 mM ATP, 5% (v/v) PEG 8000, 40 U of RiboLock RNase Inhibitor (Thermo Scientific), 20 U of T4 RNA ligase 1 and 1× T4 RNA ligase reaction buffer (NEB). Labeling reactions were carried out for 16 h at 16 °C, and RNA was subsequently purified using a Monarch RNA clean-up kit (NEB). RNA integrity was verified by denaturing urea–PAGE, with the gel imaged using an Amersham Typhoon RGB 9400 scanner (Cytiva).

### Protein preparation

*Sus scrofa* Pol II was purified as previously described[5]. The porcine and human enzymes are 99.9% identical, with only four changes within the 4,587-amino-acid sequence. TFIIF was recombinantly expressed in BL21 DE3-RIL *E. coli*. Cultures were grown to an optical density at 600 nm of 0.9 and induced by addition of 0.5 mM IPTG at 37 °C for 4 h. Cells were collected by centrifugation, resuspended in lysis buffer (50 mM HEPES-NaOH pH 7.5 at 25 °C, 350 mM NaCl, 20 mM imidazole, 10% (v/v) glycerol, 1 mM dithiothreitol (DTT)) plus protease inhibitors (1 mM PMSF, 2 mM benzamidine, 1 μM leupeptin and 2 μM pepstatin) and lysed by sonication. TFIIF was purified by affinity chromatography using a HisTrap HP column (Cytiva) equilibrated with lysis buffer. Protein was eluted with lysis buffer containing 400–500 mM imidazole, dialyzed (50 mM HEPES-NaOH pH 7.5 at 25 °C, 150 mM NaCl, 10% (v/v) glycerol, 1 mM DTT) and cleaved by 3C protease at 4 °C overnight. Cleaved protein was diluted to 100 mM NaCl and purified using a HiTrap SP column equilibrated with ion exchange buffer (50 mM HEPES-NaOH pH 7.5 at 25 °C, 10% (v/v) glycerol, 1 mM DTT) containing 100 mM NaCl. Protein was eluted using a linear gradient of 0.1–1 M NaCl in ion exchange buffer and further purified using a MonoQ (TFIIF wild type, W164A and RAP30ΔC) or a MonoS (TFIIF RAP74ΔC) column equilibrated with ion exchange buffer containing 100 mM NaCl. Protein was eluted as before and subjected to size exclusion chromatography using a Superdex 200 10/300 column (TFIIF wild type, W164A and RAP30ΔC) or a Superdex 75 10/300 column (TFIIF RAP74ΔC) (Cytiva) equilibrated with 50 mM HEPES-NaOH pH 7.5 at 25 °C, 150 mM NaCl, 10% (v/v) glycerol and 1 mM DTT. Purified TFIIF was concentrated using a Amicon Ultra-4 concentrator with a 30 kDa (TFIIF wild type, W164A and RAP30ΔC) or 10 kDa (TFIIF RAP74ΔC) cutoff (Amicon) and stored at −80 °C until use.

### Mass photometry

RNA was refolded by heating to 95 °C for 5 min followed by snap cooling on ice in 50 mM KCl binding buffer (20 mM HEPES-KOH pH 7.5 at 25 °C, 50 mM KCl, 10 μM $ZnCl_2$, 4 mM $MgCl_2$, 10 mM DTT). Pol II and refolded RNA were diluted to 1 μM in the same buffer. The complex was prepared by mixing 2.5 μl diluted Pol II with 2.5 μl diluted RNA, followed by incubation at 30 °C for 30 min. Samples were placed on ice and diluted fivefold in 50 mM KCl binding buffer. Measurements were carried out using a Refeyn Two$^{MP}$ mass photometer (Refeyn Ltd). The instrument was focused with 10 μl of buffer and 1 μl of sample was used for each measurement. Movies were collected for 1 min and processed with DiscoverMP software. Calibration of the instrument was performed before each data collection session.

### Cryo-EM sample preparation

Pol II was diluted in 150 mM NaCl polymerase buffer (5 mM HEPES-KOH pH 7.25 at 25 °C, 150 mM NaCl, 10 μM $ZnCl_2$, 10 mM DTT) and mixed with a 1.5-fold excess of refolded RNA, which had been heated to 95 °C for 5 min and snap cooled on ice in 150 mM NaCl polymerase buffer. The complex was incubated at 30 °C for 30 min, diluted to a salt concentration of 50 mM NaCl and placed on ice. The final concentration of Pol II was 400 nM.

Four microliters of sample were applied to graphene oxide-coated Quantifoil R1.2/1.3 holey carbon grids[6] that had been glow discharged for 5 s (23 mA current, $7.0 \times 10^{-1}$ mbar vacuum). Using a Vitrobot Mark IV set to 100% humidity and 4 °C, grids were blotted for 13 s and immediately plunge frozen in liquid ethane.

### Cryo-electron microscopy

Grids were screened for particle density, particle orientation and graphene oxide coverage using a Thermo Fisher Glacios transmission electron microscope (200 kV) equipped with a Falcon III direct electron detector. Grids were transferred to a Thermo Fisher Titan Krios G3i transmission electron microscope (300 kV) equipped with a Gatan K3 BioQuantum direct electron detector (energy slit width 10 eV) for data collection. Datasets were recorded using SerialEM v.3.8 and EPU v.2.11 (Pol II–Alu-RA), SerialEM v.3.9 (Pol II–Alu-LA) and EPU v.2.11 (Pol II−minimal Alu-RA). Micrographs were collected at a nominal magnification of ×105,000, corresponding to a counting mode object scale pixel size of 0.835 Å. For Pol II–Alu-LA, the dataset consisted of 16,470 micrographs collected using a defocus range of −0.2 to −2.0 μm with an electron exposure rate of 18.7 e$^-$/Å$^2$/s and an exposure time 2.14 s. The total electron exposure was 40 e$^-$/Å$^2$ distributed over 40 frames. For Pol

II–Alu-RA, the dataset consisted of 29,262 micrographs collected using a defocus range of −0.2 to −2.2 μm with an average electron exposure rate of 18.2 e⁻/Å²/s and an average exposure time of 2.2 s. The total electron exposure was 40 e⁻/Å² distributed over 40 frames. For Pol II–minimal Alu-RA, the dataset consisted of 7,588 micrographs collected using a defocus range of −0.1 to −2.2 μm with an electron exposure rate of 22.7 e⁻/Å²/s and an exposure time 1.81 s. The total electron exposure was 41 e⁻/Å² distributed over 40 frames.

Micrographs were first processed with Warp[35] and visually screened to remove micrographs with poorly visible Thon rings or multiple layers of graphene oxide. Micrographs were motion corrected and dose weighted using RELION v.3.1 (ref. 36), followed by contrast transfer function estimation using CTFFIND4 (ref. 37). Particle coordinates were selected with Topaz[38] using a very permissive threshold to ensure all particles were identified. Datasets were subsequently cleaned using a combination of 2D and 3D classification in RELION v.3.1 and 2D classification in CryoSPARC[39] (Extended Data Figs. 2–4). Final reconstructions were obtained using 3D refinement with defocus refinement, beam tilt refinement and particle polishing in RELION v.3.1 and a final binned pixel size of 1.0 Å. Local resolution was calculated in CryoSPARC using an FSC cutoff of 0.5. Densities displaying well-resolved clamp and stalk domains could be obtained after additional 3D classification with appropriate masks (Extended Data Figs. 2 and 3).

## Modeling and refinement
A model for the Pol II core lacking the stalk and clamp domains, generated from the Pol II DSIF-EC structure (Protein Data Bank (PDB) ID 5OIK)[27], was rigid body fit into the Pol II–Alu-RA map using UCSF Chimera[40]. The conformation of fork loop 2 (RBP2 487–499) was manually adjusted in Coot[41], and the resulting model was adjusted using Isolde[42]. The model was real space refined in Phenix[43] using Isolde-suggested parameters (global minimization with reference restraints and ADP refinement). The minimal Alu-RA RNA was de novo modeled into an initial Pol II–minimal Alu-RA map (before classification of the two RNA conformations) using DeepTracer[8]. Using the final classified, B-factor-sharpened minimal Alu-RA alternative conformation map, the model was adjusted in Coot[41] and the sequence register was assigned, guided by the minimum free energy RNAfold[9] secondary structure prediction generated using energy parameters rescaled to 4 °C, as energy parameters at 37 °C yielded the canonical Alu-RA conformation. Base G52 was not built, despite potential density for a base stacked on residue RBP1 Y859, due to density quality. Clamp and adjusted core Pol II models were rigid body fitted into the Pol II–minimal Alu-RA alternative conformation map, regions lacking any density in the unsharpened electron microscopy map were deleted and the combined Pol II-RNA model was further adjusted in Isolde. The model was real space refined in Phenix[43] using Isolde-suggested parameters (global minimization with reference restraints and ADP refinement). Densities and models were visualized using UCSF ChimeraX[44]. Difference densities for Pol II–Alu-LA and Pol II–Alu-RA reconstructions were visualized after subtraction of individually fitted Pol II core, clamp and stalk domains. For visualization purposes, the Pol II–minimal Alu-RA alternative conformation map was postprocessed with DeepEMhancer[45].

## Fluorescence anisotropy binding experiments
ATTO-488-labeled RNA was folded by heating to 65 °C for 5 min followed by controlled cooling to 4 °C (1 °C per 30 s) in 5 mM HEPES-NaOH pH 7.25 at 25 °C, 150 mM NaCl and 1 mM MgCl₂. Pol II–Alu RNA complexes were prepared using 4 nM RNA and increasing concentrations of Pol II (0–500 nM). Reactions were prepared in a final volume of 20 μl (5 mM HEPES-NaOH pH 7.25 at 25 °C, 150 mM NaCl, 10 μM ZnCl₂, 1 mM MgCl₂, 1 mM DTT) and incubated at 30 °C for 30 min. All experiments were performed in triplicate. Fluorescence anisotropy was measured in 384-well plates and data were collected using a Plate Reader Synergy H1-MF (Bio-TEK). Data were analyzed in GraphPad Prism v.9 using a

single site quadratic binding equation accounting for ligand depletion[46]. Values represent the mean and error bars represent the standard deviation.

## Transcription and inhibition assays
Pol II was diluted in transcription buffer (20 mM HEPES-NaOH pH 7.5 at 25 °C, 50 mM NaCl, 5 μM ZnCl₂, 4 mM MgCl₂, 4 (v/v) % glycerol, 1 mM DTT) and mixed with 5 pmol of a tailed DNA template (template strand sequence 5′-ACAAATTACTGGGAAGTCGACTAT GCAATACAGGCATCATTTGATCAAGCTCAAGTAC TTAATCATAACCATA-3′, nontemplate strand sequence 5′-TAAGTA CTTGAGCTTGATCAAATGATGCCTGTATTGCATAGTCGA CTTCCCAGTAATTTGT-3′) and bovine serum albumin (BSA, 40 μg ml⁻¹). Reactions were incubated for 10 min at 30 °C. The final concentration of Pol II was 0.12 μM. An NTP mix containing radioactive CTP (625 μM ATP, GTP, UTP; 17 μM CTP and 0.3 μM [α-32P] CTP) was added to the reaction and incubated for 10 min at 37 °C. Transcription was stopped by adding an equal volume of stop buffer (8 M urea, 20 mM EDTA), incubated for 5 min at 95 °C and snap cooled on ice. The reaction was incubated with Proteinase K (0.2 mg ml⁻¹) for 20 min at 37 °C, then heat inactivated for 5 min at 95 °C. RNA was resolved on a denaturing gel (20% acrylamide, 8 M Urea).

For inhibition assays, folded Alu RNA or RNA 151 was preincubated with Pol II for 30 min at 30 °C (protein:RNA molar ratio of 1:4). DNA template was added and transcription was carried out as described above. For reactions containing TFIIF, Pol II was preincubated with TFIIF or TFIIF mutants (Pol II:TFIIF molar ratio of 1:4) for 1 h at 30 °C, then Alu RNA was added to the reaction. Assays were carried out as described above. For detection of radioactivity, gels were exposed to a storage phosphor screen overnight at 4 °C and imaged with an Amersham Typhoon RGB 9400 scanner (Cytiva). Data were analyzed using ImageQuant TL v.10.2 analysis software (Cytiva).

## Electrophoretic mobility shift assays
RNA was folded as in the fluorescence anisotropy binding experiments. RNA (5 nM) was mixed with increasing concentrations of wild-type or mutant TFIIF (0, 15, 30 and 60 nM) in a final buffer containing 20 mM HEPES-NaOH pH 7.25 at 25 °C, 150 mM NaCl, 1 mM MgCl2, 5% (v/v) glycerol, 2 U of RiboLock RNase Inhibitor (Thermo Scientific), 40 μg ml⁻¹ BSA and 1 mM DTT. Reactions were incubated for 30 min at 30 °C, and loaded directly to a NativePAGE, 3 to 12%, Bis-Tris gel (Invitrogen). Gels were run in 1× NativePAGE running buffer (Invitrogen) at 4 °C (1 h, constant 150 V). Experiments were performed in triplicate. Labeled RNA was visualized using an Amersham Typhoon RGB 9400 scanner. Data were analyzed using ImageJ (v.154.g). Free RNA band intensities were quantified and normalized to the no protein control of the respective experiment to obtain a quantification of unbound RNA, which was subsequently plotted as fraction of bound RNA (1-unbound RNA).

## Reporting summary
Further information on research design is available in the Nature Portfolio Reporting Summary linked to this article.

# Data availability
Cryo-EM maps for Pol II–Alu-LA, Pol II–Alu-RA, Pol II–minimal-Alu-RA canonical conformation and Pol II–minimal-Alu-RA alternative conformation were deposited to the Electron Microscopy Data Bank under the accession codes EMD-18367, EMD-18375, EMD-18371 and EMD-18376. Model coordinates for the Pol II–Alu-RA polymerase core and Pol II–minimal Alu-RA complex were deposited to the PDB under the accession codes 8QEP and 8QEQ. The analysis presented here used previously published and publicly accessible model coordinates, which are available at the PDB under the accession codes 5OIK and 5IYB. Detailed plasmid maps are available on request. Source data are provided with this paper.

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

## Acknowledgements

We thank the members of the Bernecky laboratory for helpful discussions and A. Hlavata for providing Pol II for use in the fluorescence anisotropy binding assay. We thank V.-V. Hodirnau for SerialEM data collection and support with EPU data collection. We thank D. Slade (Max Perutz Laboratories and Medical University of Vienna, Vienna, Austria) for the wild-type TFIIF expression plasmid. We thank N. Thompson and R. Burgess (McArdle Laboratory for Cancer Research, University of Wisconsin-Madison, Madison, WI, USA) for the 8WG16 hybridoma cell line. We thank C. Plaschka and M. Loose for critical reading of the manuscript. This work was supported by Austrian Science Fund (FWF) grant no. P34185 (DOI 10.55776/P34185) (C.B.). The funders had no role in study design, data collection and analysis, decision to publish or preparation of the manuscript. This research was further supported by the Scientific Service Units of ISTA through resources provided by the Laboratory Support Facility, Electron Microscopy Facility, Scientific Computing and the Preclinical Facility.

## Author contributions

K.T., B.K., C.B. and A.S. performed experiments and analyzed the data. K.T., B.K. and A.S. prepared protein and RNA reagents. C.B. designed and supervised research. K.T. and C.B. processed cryo-EM data. K.T., B.K. and C.B. prepared the manuscript, with input from A.S.

## Competing interests

The authors declare no competing interests.

## Additional information

**Extended data** is available for this paper at https://doi.org/10.1038/s41594-024-01448-7.

**Correspondence and requests for materials** should be addressed to Carrie Bernecky.

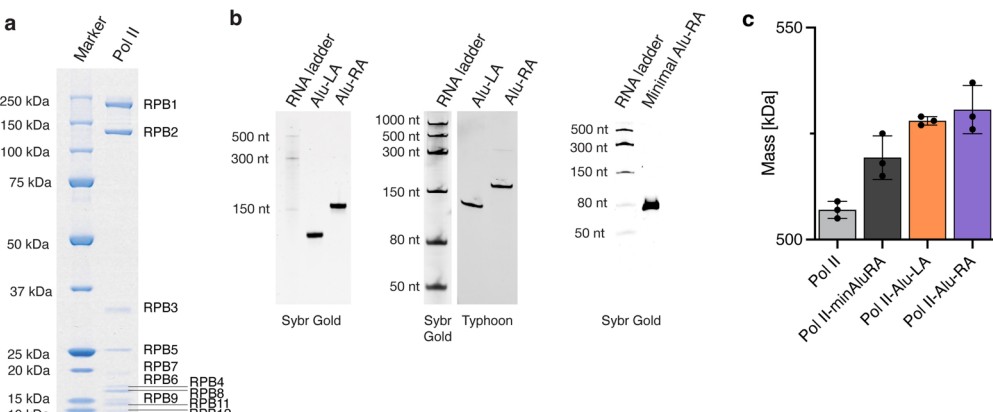

**Extended Data Fig. 1 | Preparation of Pol II-Alu RNA complexes. a**, SDS-PAGE analysis (Coomassie) of purified Pol II **b**, Denaturing urea-PAGE analysis of purified Alu RNAs. Unlabeled RNAs were visualized by SYBR Gold staining, and fluorescently labeled RNAs were visualized using a Typhoon RGB scanner.

**c**, Mass photometry analysis of Pol II-Alu complexes. The mass of the monomeric Pol II mass photometry peak was measured in technical triplicate, with each data point shown as a black circle. Bar heights represent the average and error bars represent the standard deviation from the average.

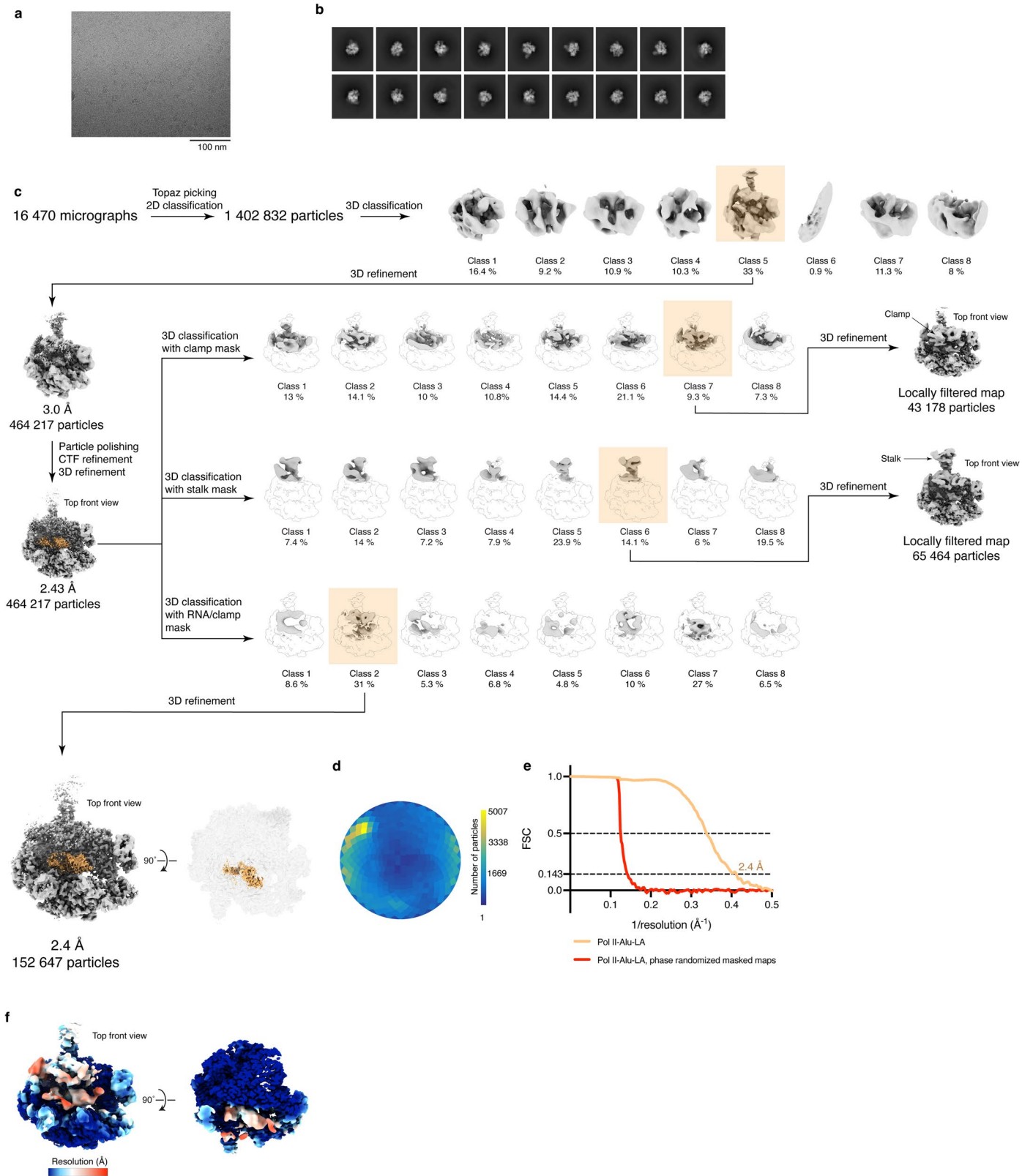

**Extended Data Fig. 2 | Cryo-EM analysis of Pol II-Alu-LA. a**, Cryo-EM micrograph representative field of view, -1.9 µm defocus. One of a total of 16,470 micrographs is shown. **b**, Representative 2D classes. **c**, Processing tree outlining the steps taken to generate the Pol II-Alu-LA reconstruction. Classes with well-resolved clamp and stalk domains are indicated. **d**, Angular distribution plot for the Pol II-Alu-LA reconstruction. **e**, Fourier shell correlation plot for the Pol II-Alu-LA reconstruction and the Pol II-Alu-LA phase-randomized masked maps. **f**, Locally-filtered, non-B-factor-sharpened Pol II-Alu-LA reconstruction colored by estimated local resolution.

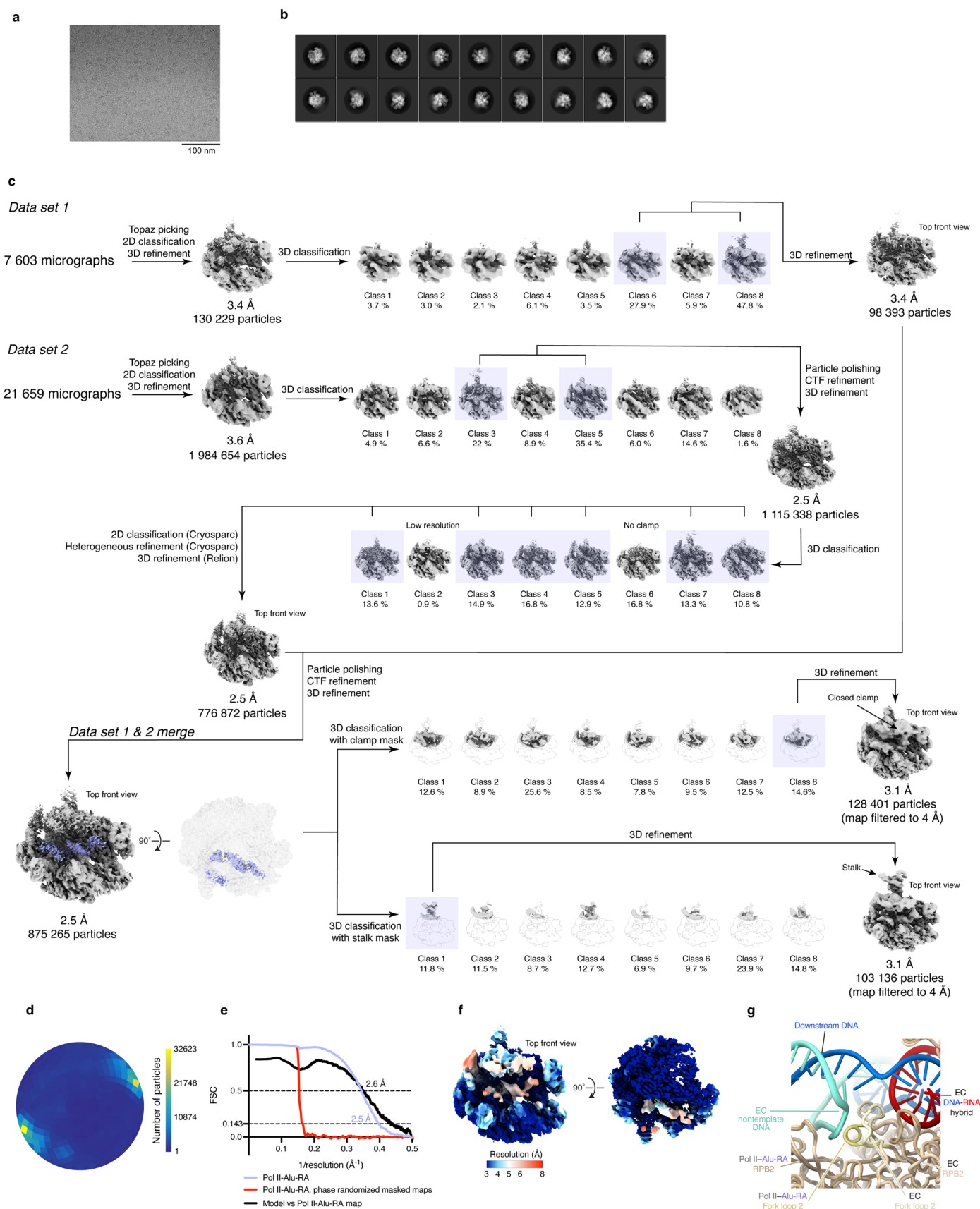

**Extended Data Fig. 3 | See next page for caption.**

**Extended Data Fig. 3 | Cryo-EM analysis of Pol II-Alu-RA. a**, Cryo-EM micrograph representative field of view, -1.6 μm defocus. One of a total of 29,262 micrographs is shown. **b**, Representative 2D classes. **c**, Processing tree outlining the steps taken to generate the Pol II-Alu-RA reconstruction. Classes with well-resolved clamp and stalk domains are indicated. **d**, Angular distribution plot for the Pol II-Alu-RA reconstruction. **e**, Fourier shell correlation plot for the Pol II-Alu-RA reconstruction, the Pol II-Alu-RA phase-randomized masked maps, and the model-versus-map correlation for the Pol II-Alu-RA reconstruction and the Pol II core model. **f**, Locally-filtered, non-B-factor-sharpened Pol II-Alu-RA reconstruction colored by estimated local resolution. **g**, Comparison of the fork loop 2 conformation in the Pol II-Alu-RA structure and a transcription elongation complex structure (PDB ID 5OIK)[27]. Coloring is as follows: Pol II-Alu-RA: RBP2 tan, fork loop 2 yellow; EC: RPB2 beige, fork loop 2 light yellow, template DNA blue, nontemplate DNA cyan, RNA red.

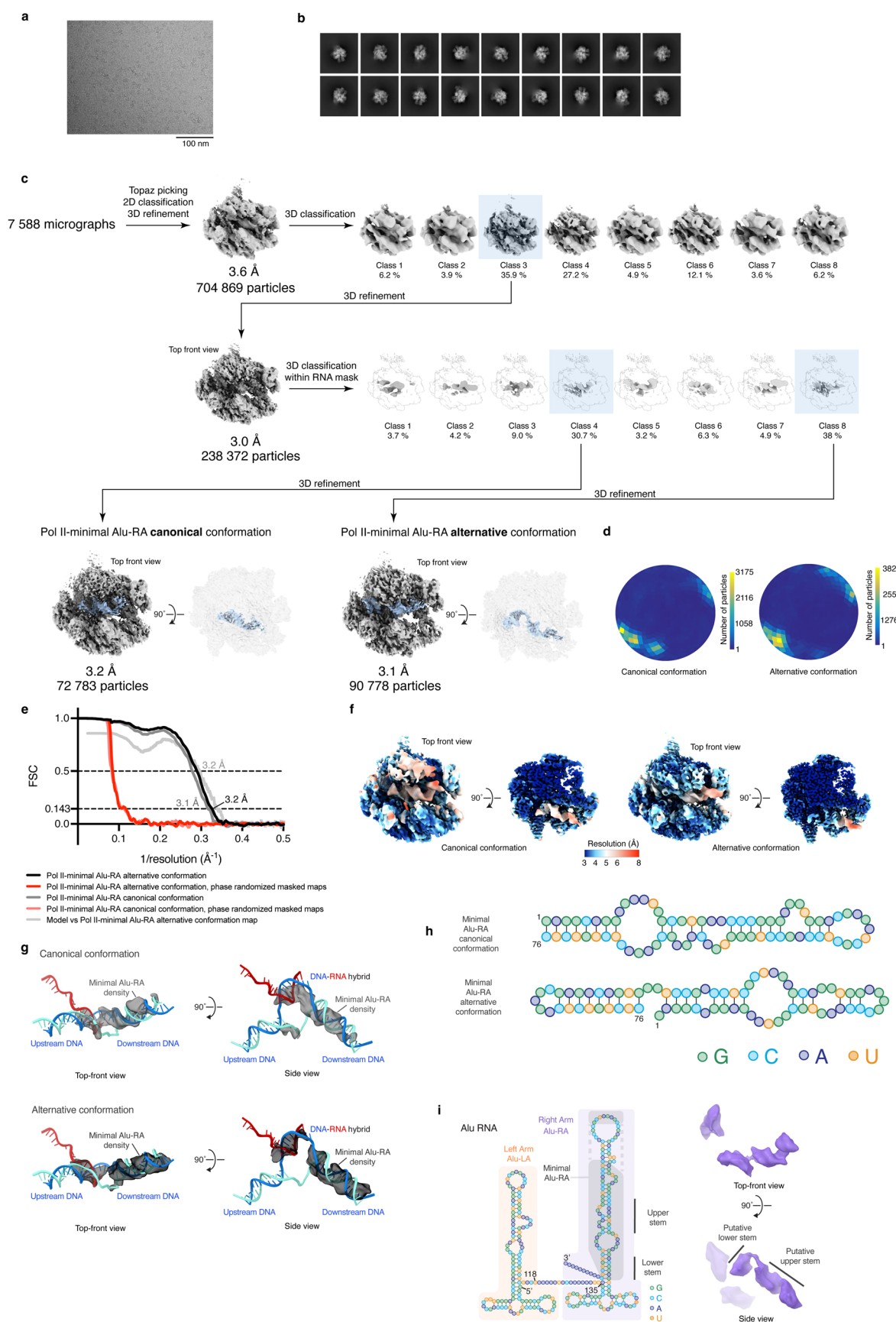

**Extended Data Fig. 4 | See next page for caption.**

**Extended Data Fig. 4 | Cryo-EM analysis of Pol II-minimal Alu-RA. a**, Cryo-EM micrograph representative field of view, -1.7 μm defocus. One of a total of 7,588 micrographs is shown. **b**, Representative 2D classes. **c**, Processing tree outlining the steps taken to generate the Pol II-minimal Alu-RA reconstructions. **d**, Angular distribution plot for the Pol II-minimal Alu-RA reconstructions. **e**, Fourier shell correlation plot for the Pol II-minimal Alu-RA reconstructions, the Pol II-minimal Alu-RA phase-randomized masked maps, and the model-versus-map correlation for the Pol II-minimal Alu-RA alternative conformation reconstruction and the model. **f**, Locally-filtered Pol II-minimal Alu-RA reconstructions colored by estimated local resolution. The Pol II-minimal Alu-RA canonical conformation reconstruction was not B-factor sharpened. **g**, Top-front and side views of 8-Å-filtered Pol II-minimal Alu-RA canonical and alternative conformation densities compared to the path of nucleic acids in a transcription elongation complex (PDB ID 5OIK)[27]. Minimal Alu-RA density is shown in semi-transparent gray. Compare to Fig. 1b,c. **h**, Schematics showing the secondary structures of the canonical and alternative conformations of minimal Alu-RA. Guanosine, green; cytidine, cyan; adenosine, blue; uridine, orange. **i**, Putative assignment of the regions of Alu-RA that bind in the DNA-RNA hybrid (putative lower stem) and the downstream DNA-binding (putative upper stem) regions of the Pol II cleft. Alu-RA difference density (purple), upper and lower stems (gray bars).

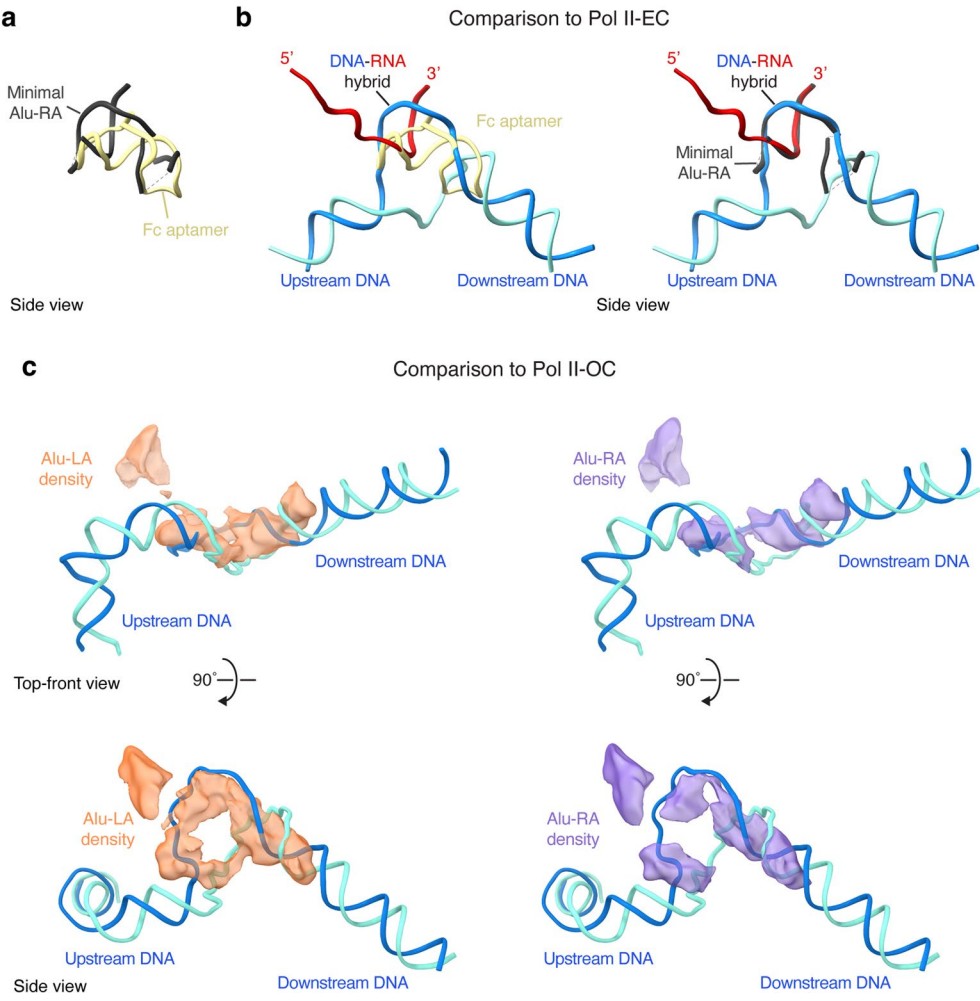

**Extended Data Fig. 5 | Comparison of Alu RNA and nucleic acids within other Pol II complexes. a**, Comparison of the paths of the nucleic acids in the Pol II-minimal Alu-RA structure and the yeast Pol II-Fc aptamer structure (PDB ID 2B63)[10] after alignment on RPB2. **b**, Comparison of the paths of Fc aptamer and minimal Alu-RA and nucleic acids in a transcription elongation complex (PDB ID 5OIK)[27]. **c**, Top-front and side views of 8-Å-filtered Alu-LA and Alu-RA densities compared to the path of nucleic acids in a transcription initiation open promoter complex (PDB ID 5IYB)[25].

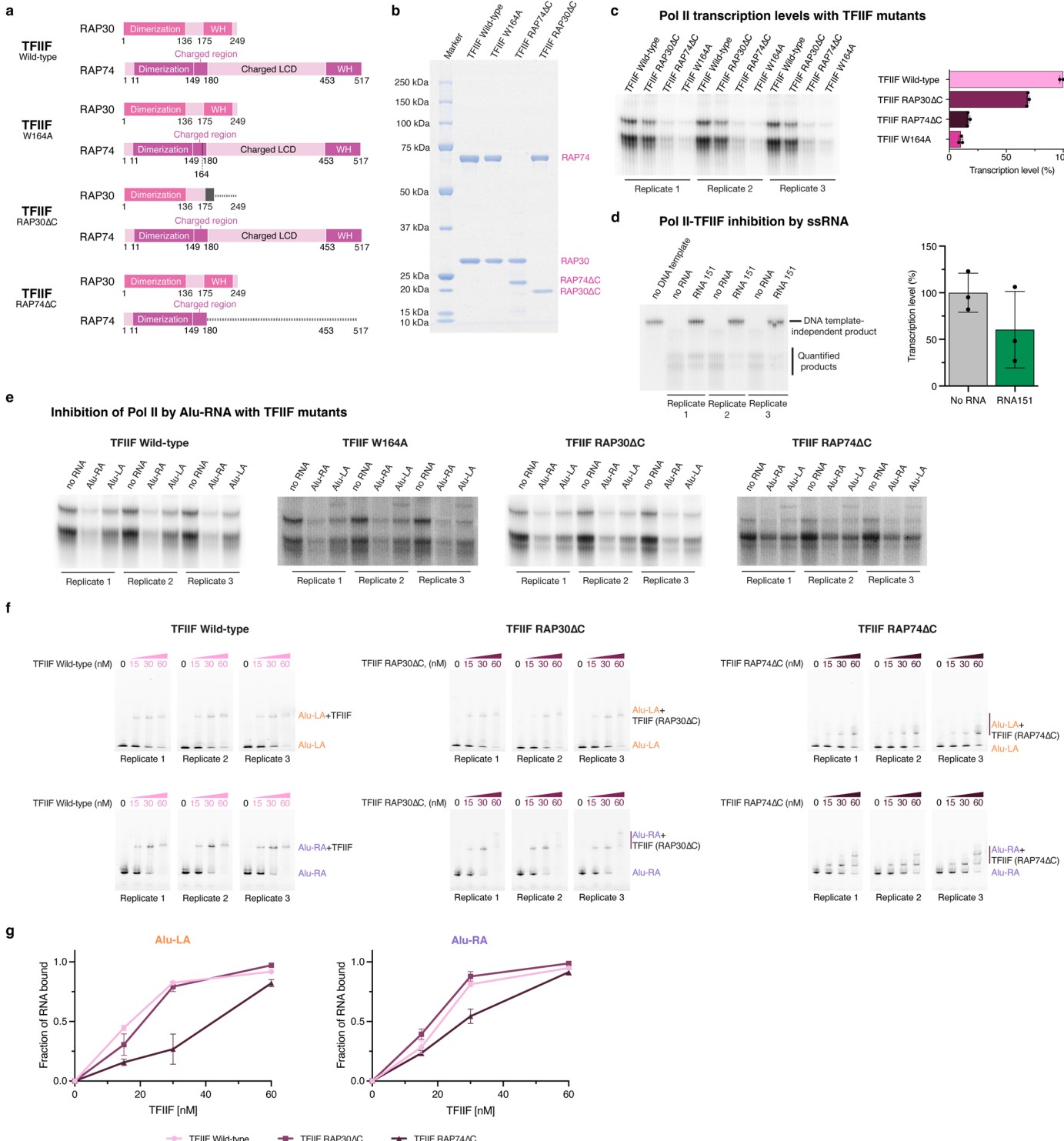

**Extended Data Fig. 6 | Preparation and activity of complexes with TFIIF and TFIIF mutants. a**, Schematic detailing the TFIIF variants used in these studies. **b**, SDS-PAGE analysis of equimolar amounts of each TFIIF variant. **c**, Transcription assays show that the TFIIF variants employed in this study have differential ability to stimulate Pol II activity, as previously described[47,48]. Left, Urea PAGE analysis and phosphorimaging; right, quantification. Data are presented as mean values +/- standard deviation, with individual data points shown as black circles. TFIIF color code as in Fig. 2. **d**, Transcription assays showing the inhibitory effect of a single-stranded RNA. Only the RNA products well separated from a DNA template-independent RNA product were quantified. Left, Urea PAGE analysis and phosphorimaging; right, quantification. Data are presented as mean values +/- standard deviation, with individual data points shown as black circles. **e**, Urea-PAGE analysis of transcription assay quantified in Fig. 2c. **f**, Electrophoretic mobility shift assay (EMSA) showing the binding of wild-type TFIIF and the indicated TFIIF mutants to Alu-LA and Alu-RA. **g**, Quantification of the fraction of bound RNA monitored by the disappearance of the free RNA band. Results represent analysis of three independently prepared reactions. Data are plotted as mean values +/- standard deviation.

# Reporting Summary

## Statistics

For all statistical analyses, confirm that the following items are present in the figure legend, table legend, main text, or Methods section.

| n/a | Confirmed | |
|---|---|---|
| ☐ | ☒ | The exact sample size (*n*) for each experimental group/condition, given as a discrete number and unit of measurement |
| ☐ | ☒ | A statement on whether measurements were taken from distinct samples or whether the same sample was measured repeatedly |
| ☐ | ☒ | The statistical test(s) used AND whether they are one- or two-sided *Only common tests should be described solely by name; describe more complex techniques in the Methods section.* |
| ☒ | ☐ | A description of all covariates tested |
| ☒ | ☐ | A description of any assumptions or corrections, such as tests of normality and adjustment for multiple comparisons |
| ☐ | ☒ | A full description of the statistical parameters including central tendency (e.g. means) or other basic estimates (e.g. regression coefficient) AND variation (e.g. standard deviation) or associated estimates of uncertainty (e.g. confidence intervals) |
| ☐ | ☒ | For null hypothesis testing, the test statistic (e.g. *F*, *t*, *r*) with confidence intervals, effect sizes, degrees of freedom and *P* value noted *Give P values as exact values whenever suitable.* |
| ☒ | ☐ | For Bayesian analysis, information on the choice of priors and Markov chain Monte Carlo settings |
| ☒ | ☐ | For hierarchical and complex designs, identification of the appropriate level for tests and full reporting of outcomes |
| ☒ | ☐ | Estimates of effect sizes (e.g. Cohen's *d*, Pearson's *r*), indicating how they were calculated |

*Our web collection on statistics for biologists contains articles on many of the points above.*

## Software and code

Policy information about availability of computer code

| Data collection | SerialEM 3.8, SerialEM 3.9, EPU 2.11 |
|---|---|
| Data analysis | Warp 1.0.9, RELION 3.1, CTFFIND4, Topaz 0.2.5, CryoSPARC 3.2, Coot 0.9.6, UCSF Chimera 1.16, UCSF ChimeraX 1.6, Isolde 1.6, Phenix 1.20.1, DeepEMhancer 20210511, GraphPad Prism 9, ImageQuant TL 10.2, ImageJ (version 154.g), DiscoverMP 2.3 |

For manuscripts utilizing custom algorithms or software that are central to the research but not yet described in published literature, software must be made available to editors and reviewers. We strongly encourage code deposition in a community repository (e.g. GitHub). See the Nature Portfolio guidelines for submitting code & software for further information.

## Data

Policy information about availability of data

All manuscripts must include a data availability statement. This statement should provide the following information, where applicable:
- Accession codes, unique identifiers, or web links for publicly available datasets
- A description of any restrictions on data availability
- For clinical datasets or third party data, please ensure that the statement adheres to our policy

Cryo-EM maps for Pol II-Alu-LA, Pol II-Alu-RA, Pol II-minimal-Alu-RA canonical conformation, and Pol II-minimal-Alu-RA alternative conformation were deposited to the EM Data Bank under the accession codes EMD-18367, EMD-18375, EMD- 18371, and EMD-18376. Model coordinates for the Pol II-AluRA polymerase core and

Pol II-minimal Alu-RA complex were deposited to the PDBe under the accession codes 8QEP and 8QEQ. Previously published and publicly accessible model coordinates are available from the PDBe under the accession codes 5OIK and 5IYB.

# Research involving human participants, their data, or biological material

Policy information about studies with <u>human participants or human data</u>. See also policy information about <u>sex, gender (identity/presentation), and sexual orientation</u> and <u>race, ethnicity and racism</u>.

| | |
|---|---|
| Reporting on sex and gender | n/a |
| Reporting on race, ethnicity, or other socially relevant groupings | n/a |
| Population characteristics | n/a |
| Recruitment | n/a |
| Ethics oversight | n/a |

Note that full information on the approval of the study protocol must also be provided in the manuscript.

# Field-specific reporting

Please select the one below that is the best fit for your research. If you are not sure, read the appropriate sections before making your selection.

☒ Life sciences          ☐ Behavioural & social sciences          ☐ Ecological, evolutionary & environmental sciences

For a reference copy of the document with all sections, see nature.com/documents/nr-reporting-summary-flat.pdf

# Life sciences study design

All studies must disclose on these points even when the disclosure is negative.

| | |
|---|---|
| Sample size | No sample size was calculation was performed. Biochemical assays were performed in triplicate to allow statistical analysis of the data. Number of collected micrographs was chosen based on expected heterogeneity and flexibility of the analyzed complex and the availability of microscope resources. |
| Data exclusions | No data were excluded. |
| Replication | Sample preparation and analysis were reproducible. For each sample, one small and one large data set were collected. Initial reconstructions from small data sets resembled the analysis of the full data sets. |
| Randomization | During 3D refinement, cryo-EM data were split randomly into two halves for gold-standard FSC determination. Biochemical experiments were not randomized, but contained appropriate controls. |
| Blinding | Blinding is not relevant or feasible for these types of experiments. The biochemical and structural experiments presented here are rigorously quality controlled through other criteria and mechanisms as accepted within the field. |

# Reporting for specific materials, systems and methods

We require information from authors about some types of materials, experimental systems and methods used in many studies. Here, indicate whether each material, system or method listed is relevant to your study. If you are not sure if a list item applies to your research, read the appropriate section before selecting a response.

## Materials & experimental systems

| n/a | Involved in the study |
|---|---|
| ☐ | ☒ Antibodies |
| ☒ | ☐ Eukaryotic cell lines |
| ☒ | ☐ Palaeontology and archaeology |
| ☐ | ☒ Animals and other organisms |
| ☒ | ☐ Clinical data |
| ☒ | ☐ Dual use research of concern |
| ☒ | ☐ Plants |

## Methods

| n/a | Involved in the study |
|---|---|
| ☒ | ☐ ChIP-seq |
| ☒ | ☐ Flow cytometry |
| ☒ | ☐ MRI-based neuroimaging |

## Antibodies

| | |
|---|---|
| Antibodies used | 8WG16 (hybridoma line provided by N. Thompson and R. Burgess) |
| Validation | This antibody was used to purify RNA polymerase II. |

## Animals and other research organisms

Policy information about studies involving animals; ARRIVE guidelines recommended for reporting animal research, and Sex and Gender in Research

| | |
|---|---|
| Laboratory animals | This study did not involve laboratory animals. |
| Wild animals | This study did not involve wild animals. |
| Reporting on sex | n/a |
| Field-collected samples | This study did not involve field samples |
| Ethics oversight | This project does not raise ethical issues. Pig thymus (used for protein purification) is sourced from animals approved for food consumption through an officially approved facility in Sieghartskirchen, Lower Austria. |

Note that full information on the approval of the study protocol must also be provided in the manuscript.

