## [Peer Review File · Nature Structural & Molecular Biology]

Mechanism of mammalian transcriptional repression by noncoding RNA

Corresponding Author: Dr Carrie Bernecky

Version 0:

Decision Letter:

10th Oct 2023

Dear Dr. Bernecky,

Thank you again for submitting your manuscript "Mechanism of mammalian transcriptional repression by noncoding RNA". I apologise for the delay in responding, which resulted from the difficulty in obtaining suitable referee reports. Nevertheless, we now have comments (below) from the 2 reviewers who evaluated your paper. In light of these reports, we remain interested in your study and would like to see your response to the comments of the referees, in the form of a revised manuscript.

You will see that the referees appreciate the structural and biochemical data, and view their interpretation and implications as potentially interesting. However, both experts raise very important concerns and questions which need to be adequately addressed in a revised manuscript. More specifically, both experts wonder if the structural data can explain the functional differences of the two Alu segments and request additional information with respect to how different the "canonical" and "alternative" conformations are in the high-resolution structure of the truncated Alu-RA. Moreover, both referees raise technical concerns and/or ask for technical clarifications which must be addressed in their entirety. Reviewer #1 additionally focuses on the need for further biochemical data to support the presented findings and reviewer #2 poses several mechanistic questions and challenges to the proposed hypotheses that need to be convincingly addressed.

Please be sure to address/respond to all concerns of the referees in full in a point-by-point response and highlight all changes in the revised manuscript text file.

We appreciate the requested revisions are extensive. We thus expect to see your revised manuscript within 6 months. If you cannot send it within this time, please let us know. We will be happy to consider your revision as long as nothing similar has been accepted for publication at NSMB or published elsewhere. Should your manuscript be substantially delayed without notifying us in advance and your article is eventually published, the received date would be that of the revised, not the original, version.

Reporting Summary:
<https://www.nature.com/documents/nr-reporting-summary.pdf>

We require deposition of coordinates (and, in the case of crystal structures, structure factors) into the Protein Data Bank with the designation of immediate release upon publication (HPUB). Electron microscopy-derived density maps and coordinate data must be deposited in EMDB and released upon publication. Deposition and immediate release of NMR chemical shift assignments are highly encouraged. Deposition of deep sequencing and microarray data is mandatory, and the datasets must be released prior to or upon publication. To avoid delays in publication, dataset accession numbers must be supplied with the final accepted manuscript and appropriate release dates must be indicated at the galley proof stage. Please find the complete NRG policies on data availability at <http://www.nature.com/authors/policies/availability.html>.

Link Redacted

Sincerely,

Dimitris Typas
Associate Editor
Nature Structural & Molecular Biology
ORCID: 0000-0002-8737-1319

Reviewers' Comments:

Reviewer #1:

Remarks to the Author:

In this manuscript, Tlučková et al. present cryo-EM structures of mammalian PolIII bound with Alu RNA fragments (Alu-LA and Alu-RA), which shows that these RNAs are bound to the PolIII cleft. The authors also report the high-resolution, alternative conformation of the RNA fragment bound to the PolIII cleft, by utilizing the truncated version of the Alu-RA RNA. Finally, the authors perform competition/transcription assays using several TFIIF deletion mutants, to analyze how TFIIF domains can affect the repressive activity of Alu RNA fragments. While these complex structures of improved quality are potentially interesting, there are several concerns that should be addressed, and more experiments might be preferred to explain the structural basis of, for example, why Alu-RA is repressive but Alu-LA is not.

Major comments:

>Alu-LA and Alu-RA structures

While the resolution is limited, it would be more informative if authors could assign blobs to, for example, specific segments

of RNA, possibly by secondary structure predictions and combining biochemical assays.

>the high resolution alternative structure of the truncated Alu-RA

Please describe more about the structural difference between the canonical and the alternative conformation, with a figure. Also, side by side comparison of the RNA 3D structure and the secondary structure(as in fig1a) might be helpful for understanding.

If the alternative conformation is visibly different from the canonical, it could be inadequate to draw conclusion about Alu RNAs from this high-resolution structure.

>"We conclude that this region (*loosely structured region) ~~~~~ allows the RNA to bend and adopt a conformation matching that of the elongation complex DNA-RNA hybrid and downstream DNA"

Please provide biochemical assays that can describe the importance of this region and its bending. The reviewer thinks it is difficult to conclude this from the structure alone, if there is no tangible structural difference between Alu-LA (without LSR) and Alu-RA (with LSR). Also, it was not clear from the manuscript which part of the cryo-em structure(density) correspond to this loosely structured region and which part (which residue) is actually bending.

>"We propose that without the loose region of secondary structure shown to be important for Alu-RA transcription inhibition, Alu-LA is less stable in the bent conformation required to bind in an EC-like conformation, and therefore more labile and more susceptible to dissociation from Pol II bound to TFIIIF"

If this is true, why Alu-LA has higher affinity with PolII (as shown in Fig3a).

Note that while Fig3a was measured without TFIIIF, the requirements for the bent conformation should not depend on TFIIIF, as structures are determined without TFIIIF.

>Assays in Fig3

The authors demonstrate that Koff is affected by TFIIIF variants. However, is it safe to assume that Kon(or Kd) is not affected? Is transcription repression related only with Koff?(and not Kon or Kd?)

Ideally, I would prefer using non-structured RNA of similar length as a negative control(in addition to the "no RNA") as there could be non-specific effect of adding polynucleotides.

>"We observed similarly high affinities for Alu-LA and Alu-RA when investigated in the presence of 150 mM monovalent salt, with the non-repressive Alu-LA binding with slightly higher affinity (Fig. 3a)."

Does this mean that these assays were performed with 150mM salts?

In the methods section, 50 mM KCl (or NaCl) buffers are listed.

Please clarify.

>Cryo-EM sample preparation

The authors seem to use zero Mg²⁺ buffer for the structural analysis.

This reviewer is not completely sure if this is ok-ish, because Mg²⁺ can be important for RNA structures and/or RNA-protein interactions, and the authors include 4mM Mg²⁺ for assays.

Also, the reviewer wonders if the active site Mg²⁺ depicted in Fig2d was observed or not.

>Cryo-EM data processing

The authors performed masked classification with RNA(/clamp) masks for the Alu-LA and minimal Alu-RA datasets.

However, no such step seemingly exists for the Alu-RA dataset.

This reviewer wonders why this is the case, because consistent data processing is usually preferred if they want to compare structures from different datasets.

This reviewer also wonders if running more 3D classifications could help improve the RNA density of the Alu-RA dataset, as there are more than 800k particles in the final reconstruction.

Minor comments:

>We solved the structure of the Pol II-minimal Alu-RA complex to a nominal resolution of 3.1 Å, and observed that the quality of the RNA density was greatly improved (Fig. 2a)

This statement could be a bit misleading, as the "alternative conformation" is shown in fig2a before it is described in the next sentence. Also because it looks like the quality improvement is only in the alternative conformation, and not in the canonical conformation.

> Altogether, despite Alu RNAs adopting an EC-like conformation, the Pol II enzyme was observed in an inactive state,

consistent with the repressive activity of Alu RNA.

This reviewer doubts if the fork-loop conformation is *consistent with* the repressive activity of Alu RNA.
(This reviewer assumes that if Alu RNA binds the cleft, the polymerase can not bind the promoter or "correct" DNA, and this is why transcription is repressed. If so, the loop conformation does not seem to be relevant for the repression, unless PolIII is extending the Alu RNA.)

>Purified RNA was refolded by heating to 95°C for 5 min in buffer as specified

Please specify the buffer composition.

(Possibly the authors used 150mM NaCl buffer for cryo-em, and 50mM KCl buffer for assays?, but please make it clearer.)

>S. scrofa Pol II was purified as previously described

Please specify the composition of the Pol II storage buffer.

Note that the reviewer is interested if they have residual Mg²⁺ somewhere.

>HEPES (many times)

HEPES-NaOH or HEPES-KOH ?

>Pol II was preincubated with TFIIIF or TFIIIF mutants (protein:RNA molar ratio of 1:4)

possibly, PolIII:TFIIIF molar ratio

>, then Alu RNA was added RNA to the reaction

typo?

>Cryo-EM stats

Please add phase randomized FSCs to extended figures.

Please add number of micrographs and dose rate(e-/s/px) to Table1.

Reviewer #2:

Remarks to the Author:

The brief communication by Tlučková et al investigates the molecular mechanism of transcriptional inhibition by Alu RNA. The study is an extension of the original work by Goodrich and coworkers who identified this RNA as contributing to the inhibition of transcription initiation during heat shock and characterized the importance of smaller RNA segments and segment features in the repression mechanism using in vitro transcriptional reconstitution studies. Tlučková et al now aim to place that work and further biochemical characterization into a structural framework of PolIII-Alu-RNA interactions and how they compete with normal nucleic acid engagement in transcribing Pol II complexes. The work includes both cryo-EM studies with a number of different ALU RNA constructs, inspired by Goodrich's work, as well as biochemical and transcriptional assays. The structural work extends that published 10 years ago at much lower resolution, which already concluded the RNA was occupying the cleft of Pol II where the template DNA and the RNA-DNA hybrid should be located during initiation and elongation of Pol II.

I find the present work informative and leading to a refinement of the original models, including more quantitative data and higher resolution structures that now can try to place the biochemical findings into an structural framework that better explains them. I would support publication if the authors were to consider certain changes in their discussion and in the presentation of their data that could make the data presented and the models proposed more compelling.

The resolution on the Alu-RNA for the larger constructs is much lower than for Pol II itself. This could be because under the conditions used the occupancy of the RNA is low, or because there is heterogeneity in the RNA mode of binding (e.g. multiple registers with respect to the protein), or due to the flexibility of the RNA itself. The image analysis pipeline describes the process of 3D classification using a mask for the clamp and RNA region, but it is not clear what the findings were. For example, what were the difference between the two major classes, 2 and 7? Were the other "empty" classes with no RNA? In any case, even after this sorting, variability still reduce the resolution of the RNA, indicating either that the contacts with the protein are not extensive or that there are multiple modes of interaction. Could this be actually important for Alu-RNA inhibition?

Can the authors comment a bit more on the differences (or not) of the structures for the two Alu segments, given that they do have different activities?

The authors refer briefly to the difference in conformation of the Fork2 loop in their structure of PolIII-Alu-RA versus that in the elongating complex. Unfortunately, the corresponding figure is tiny and buried in a supplemental figure. Can they show whether the change in the Alu-containing structure is not due to contact with the Alu RNA, rather than simply due to lack of contact with a non-template strand? In that paragraph the authors state "Altogether, despite Alu RNAs adopting an EC-like conformation, the Pol II enzyme was observed in an inactive state, consistent with the repressive activity of Alu RNA." That conformation seem irrelevant to the repression process compared with the occupancy of the critical binding sites for the DNA and DNA-RNAs hybrid by the Alu-RNA!! Are the authors implying that the Alu RNA captures an inhibited state? That somehow that state will be relevant for competing off the inhibitory RNA? I fail to see the significance.

When working with the minimal domain, it would be informative to show more clearly how different, other than in resolution, are RNA densities in the structure for the "canonical" and "alternative" conformation. Could they be shown side by side, or

even superimposed, maybe by themselves and with the model of the nucleic acid in the EC? (for example, do the equivalent of Sup Fig 4g for the alternative, right next to that panel).

Can the authors speculate how the RNA could “switch” between one state and the other? It seems that the canonical minimal state shows less density than the full Alu-RA. Can the authors use that to speculate which regions of the RNA density correspond to what part of the molecule? Also, are the authors sure that the alternative conformation was not present for the RA or RL structures? Did they look for it among the discarded particles in those data sets?

I am not sure of the intent and value of the speculation in lines 105 to 115, irrespective of the mimicry of one of the conformers of a minimal Alu-RNA to the DNA-RNA hybrid. The original work of Goodrich established that Alu can only inhibit if the PolII-Alu-RNA complex is formed previous to Pol II recruitment to the promoter (presumably the PIC), at which point the loading of Pol II onto the promoter takes place and repression does not happen.

The authors try to address important functional questions, such as the fact that Alu-LA is more labile than Alu-RA in the presence of TFIIIF, in spite of both forming similar structures, via biochemical assays. They found that TFIIIF markedly reduced the half-life of Alu-LA on Pol II, with little effect on that of Alu-RA. The authors then postulate that Alu-LA is less stable in the bent conformation required to bind in an EC-like conformation, and therefore more labile and more susceptible to dissociation from Pol II bound to TFIIIF. That implies a shorter half life even in the absence of TFIIIF than that of Alu-RA, which they showed is not the case. Whatever the mechanism, it has to be one that manifest itself only in the capacity of TFIIIF to actively remove one construct versus the other. Thus, the authors need to come out with a better hypothesis of why the additional flexible element in Alu-RA makes it less susceptible to be competed off by TFIIIF.

The authors indicate that TFIIIF would not sterically clash with the resolved RNA elements. Unless their density can account for 100% of the nucleotides present in the MINIMAL construct, this argument cannot exclude steric hindrance with elements that are not visible, and in particular, with elements that are different between Alu-LA and -RA. The authors carry out transcription repression assays in the presence of TFIIIF, using both wild type and mutants. I find the explanation for the results using the TFIIIF (RAP74DC) mutant highly confusing. An important question to be addressed was why, in spite of their similar binding, only the Alu-RA half can inhibit transcription in the presence of TFIIIF? Have their data really explained this behavior?

Transcriptional repression involves Alu RNA incorporation with Pol II into stable complexes at promoters while Alu RNA does not repress transcription after Pol II has formed preinitiation complexes. These observations led Goodrich and coauthors to suggest that once Pol II is engaged with promoter DNA it is resistant to repression by Alu RNA. This can now be explained by the present work and the authors should emphasize it.

On the other hand, something that the authors do not mention, perhaps because their work cannot address it, is that two distinct regions of Alu RNA that function to mediate transcriptional repression are not required for Pol II binding (again in reference 2). In other words, the binding is not sufficient for repression. These regions, maybe not visible in EM structures, may be involved in precluding Pol II engagement with the PIC, thus avoiding competition of promoter DNA for Alu RNA. Maybe the authors have a better speculation to make in the context of their work?

Version 1:

Decision Letter:

Our ref: NSMB-BC48182A

3rd Aug 2024

Dear Dr. Bernecky,

Thank you for submitting your revised manuscript "Mechanism of mammalian transcriptional repression by noncoding RNA" (NSMB-BC48182A). It has now been seen by the original two referees and their comments are below. The reviewers find that the paper has improved in revision, and therefore we are ready to accept it in principle in Nature Structural & Molecular Biology, pending revisions to satisfy the referees' final requests and to comply with our editorial and formatting guidelines.

We are now performing detailed checks on your paper and will send you a checklist detailing our editorial and formatting requirements in approximately 4 weeks. Please do not upload the final materials and make any revisions until you receive this additional information from us.

To facilitate our work at this stage, it is important that we have a copy of the main text as a word file. If you could please send along a word version of this file as soon as possible, we would greatly appreciate it; please make sure to copy the NSMB account (cc'ed above).

Sincerely,

Dimitris Typas
Senior Editor
Nature Structural & Molecular Biology
ORCID: 0000-0002-8737-1319

Reviewer #1 (Remarks to the Author):

In this revised manuscript, the authors have made substantial changes to the text. The manuscript is generally improved, with less speculations, better readability and better presentations of data. Also, the authors included a new binding assay of TFIIIF and Alu RNA, trying to conclude that the higher binding affinity of THIIIF RAP74dC to Alu-LA (compared to the affinity to Alu-RA), is related to the functional difference between Alu-LA/RA .

Comments:

>Specifically, in Extended Data Fig. 6f and g, we show that TFIIIF binds directly to Alu-LA and Alu-RA and that interactions between Alu RNAs and TFIIIF are reduced for the TFIIIF (RAP74DC) mutant, particularly for Alu-LA. We now propose that direct TFIIIF-RNA binding may be important for relieving Alu RNA repression.

This reviewer is rather cautious about this statement, because

1. It is hard to judge from the graphs(ExFig6g) if the difference between LA/RA is statistically significant.
2. Even if it is significant, the difference does not look very large (about 2-fold affinity difference?)
3. At 60nM concentrations, there are multiple bands of complexes, which suggests that multiple TFIIIF molecules can bind to the RNA(ExFig6f). Also, the amount of upper band looks very different between TFIIIF variants, or between Alu-LA/RA. As transcription assays are performed with much higher concentrations (120nM Pol2-480nM TFIIIF-480nM RNA?), the actual situation could be much more complicated.

This reviewer is satisfied with other responses.

Reviewer #2 (Remarks to the Author):

I find that the authors have made a good attempt at answering the reviewers comments and that the paper is improved enough to warrant publication

Version 2:

Decision Letter:

8th Nov 2024

Dear Dr. Bernecky,

We are now happy to accept your revised paper "Mechanism of mammalian transcriptional repression by noncoding RNA" for publication as a Brief Communication in Nature Structural & Molecular Biology.

Your paper will be published online soon after we receive proof corrections and will appear in print in the next available issue. You can find out your date of online publication by contacting the production team shortly after sending your proof corrections.

Please note that *Nature Structural & Molecular Biology* is a Transformative Journal (TJ). Authors may publish their research with us through the traditional subscription access route or make their paper immediately open access through payment of an article-processing charge (APC). Authors will not be required to make a final decision about access to their article until it has been accepted. [Find out more about Transformative Journals](https://www.springernature.com/gp/open-research/transformative-journals)

Sincerely,

Dimitris Typas
Senior Editor
Nature Structural & Molecular Biology
ORCID: 0000-0002-8737-1319

List of responses to reviewer comments

“Mechanism of mammalian transcriptional repression by noncoding RNA”
(Katarína Tlučková, Beata Kaczmarek, Anita Salmazo, and Carrie Bernecky)

NSMB Manuscript NSMB-BC48182

Responses are in *italics*.

Reviewer comments:

Reviewer #1:

Remarks to the Author:

In this manuscript, Tlučková et al. present cryo-EM structures of mammalian PolIII bound with Alu RNA fragments (Alu-LA and Alu-RA), which shows that these RNAs are bound to the PolIII cleft. The authors also report the high-resolution, alternative conformation of the RNA fragment bound to the PolIII cleft, by utilizing the truncated version of the Alu-RA RNA. Finally, the authors perform competition/transcription assays using several TFIIF deletion mutants, to analyze how TFIIF domains can affect the repressive activity of Alu RNA fragments. While these complex structures of improved quality are potentially interesting, there are several concerns that should be addressed, and more experiments might be preferred to explain the structural basis of, for example, why Alu-RA is repressive but Alu-LA is not.

We thank the reviewer for their constructive comments, which have helped us to improve our manuscript. In the revision, we have added substantial new experimental data and have rephrased significant parts of the manuscript text, while also editing for conciseness, in line with the Brief Communication format.

Major comments:

>Alu-LA and Alu-RA structures

While the resolution is limited, it would be more informative if authors could assign blobs to, for example, specific segments of RNA, possibly by secondary structure predictions and combining biochemical assays.

Thank you for the comment. The Pol II-canonical minimal Alu-RA cryo-EM density that we observed can account for the majority of the minimal Alu-RA RNA sequence (shown in Response Fig. 1 below). Based on secondary structure prediction, the Pol II-canonical minimal Alu-RA cryo-EM density, and the Pol II-Alu-RA density, we now propose an assignment for the regions of Alu-RA that bind in the DNA-RNA hybrid and the downstream DNA-binding regions of the Pol II cleft (Extended Data Fig. 4i).

Response Figure 1. Speculative, complete model of the minimal Alu-RA canonical conformation and the observed density. The model of 10-subunit Pol II is shown in the left panel for orientation. The canonical conformation difference density, filtered to 8 Å (dust hidden) is shown in transparent gray. The canonical conformation model of minimal Alu-RA was generated by fitting the farfar2-generated lowest energy structure (Lyskov et al PLoS One. 2013) into the density.

We note, however, that the Alu-LA and Alu-RA tertiary structure predictions are variable (RNacomposer, farfar2, and AlphaFold3) and that we could not resolve additional RNA domains within our cryo-EM structures. We therefore cannot unambiguously assign further Alu regions to the EM density for the RNA. Although it is plausible from our modeling that the SRP-binding regions of Alu-LA and Alu-RA (lower horizontal stems depicted in Fig. 1a) could contribute to the “blobs” in the upstream and RNA exit regions (likely adopting many conformations across many single molecules), we have decided to not include this speculation in the manuscript to remain conservative. We hope the referee agrees with these changes.

>the high resolution alternative structure of the truncated Alu-RA

Please describe more about the structural difference between the canonical and the alternative conformation, with a figure. Also, side by side comparison of the RNA 3D structure and the secondary structure(as in fig1a) might be helpful for understanding.

If the alternative conformation is visibly different from the canonical, it could be inadequate to draw conclusion about Alu RNAs from this high-resolution structure.

We thank the reviewer for this comment. We now compare the alternative and canonical conformations in the revised Extended Data Fig. 4g, and the secondary structures are depicted side-by-side in Extended Data Fig. 4h.

Although RNAfold secondary structure prediction of full-length Alu RNA revealed the possibility of an alternative-like conformation, it is unknown whether this would exist within cells, particularly in light of potential kinetic effects on folding. However, we consider this unexpected result of the alternative conformation extremely informative for understanding mechanisms of Pol II regulation. This alternative RNA species is able to bind Pol II in a stable manner and folds into an EC-like conformation, which binds in the RNA-DNA hybrid-binding and downstream-DNA binding regions of Pol II. This observation of an additional EC-like RNA conformation supports our key conclusion that mammalian Pol II is efficiently inhibited by RNA that is able to mimic EC nucleic acids rather than promoter DNA nucleic acids.

>"We conclude that this region (*loosely structured region) ~~~~~ allows the RNA to bend and adopt a conformation matching that of the elongation complex DNA-RNA hybrid and downstream DNA"

Please provide biochemical assays that can describe the importance of this region and its

bending.

The reviewer thinks it is difficult to conclude this from the structure alone, if there is no tangible structural difference between Alu-LA (without LSR) and Alu-RA (with LSR). Also, it was not clear from the manuscript which part of the cryo-em structure (density) correspond to this loosely structured region and which part (which residue) is actually bending.

Thank you for the comment. We have now removed the text mentioning RNA bending from the manuscript. We instead focus on the key result, that an EC-like conformation rather than an open-promoter-like conformation (as was previously proposed) is adopted for the RNA-based repression of Pol II.

However, the importance of this region (nucleotides 193-209 and 257-272) and its loosely base-paired character has been shown in previous mutagenesis experiments (Mariner et al. 2008). In summary, in the previous work, it was shown that deletion of this region or its mutation to a fully base paired stem abolished the repressive ability of Alu-RA in an in vitro transcription assay. Furthermore, deletion of the base-paired regions (leading to a region predicted to be completely single stranded) also reduced the repressive activity of Alu-RA. Our structural data is fully consistent with these mutagenesis experiments, which would suggest that the two duplex stems connected by linker nucleotides may help to accommodate the RNA in the Pol II cleft in its EC-like conformation. We note that the role of base pairing within the Alu-LA stem has not been directly investigated, and that Alu-LA contains a smaller predicted mismatched region and a bulge, which could be less effective as a downstream DNA mimic than the corresponding “loose” region of Alu-LA (for visualization, compare the Alu-LA and Alu-RA schematics shown in Extended Data Fig. 4i).

>"We propose that without the loose region of secondary structure shown to be important for Alu-RA transcription inhibition, Alu-LA is less stable in the bent conformation required to bind in an EC-like conformation, and therefore more labile and more susceptible to dissociation from Pol II bound to TFIIF"

If this is true, why Alu-LA has higher affinity with PolII (as shown in Fig3a).

Note that while Fig3a was measured without TFIIF, the requirements for the bent conformation should not depend on TFIIF, as structures are determined without TFIIF.

We originally provided a speculative explanation for how the loose region of Alu-RA may confer resistance to TFIIF. During the revision period, we have carried out new RNAfold secondary structure predictions for the previously described RNA mutants, and observed that for the 4-nt-longer minimal Alu-RA construct described and tested in Mariner et al. 2018, only the alternative conformation is predicted. Thus, we cannot conclude that this region alone, in its canonical conformation, is sufficient to confer resistance to TFIIF. To be conservative in our claims, we have removed this speculation from the manuscript. Notably, these results emphasize the challenge of generating RNA mutants without altering the global fold.

In addition, we have repeated the Alu-LA and Alu-RA binding assay using an optimized fluorescence labeling strategy and slightly altered buffer conditions (including 1 mM Mg²⁺, in line with the estimated 0.5-1.2 mM intracellular free Mg²⁺ (de Baaij JH, et al. Physiol Rev.

2015). This experiment is shown in the revised Fig. 3a. Under these conditions, Alu-LA and Alu-RA display indistinguishable affinities towards Pol II alone.

>Assays in Fig3

The authors demonstrate that K_{off} is affected by TFIIIF variants. However, is it safe to assume that K_{on} (or K_d) is not affected? Is transcription repression related only with K_{off} ?(and not K_{on} or K_d ?)

We apologize for the lack of clarity. We did not mean to imply that the K_d was not affected. Rather, we meant to convey that we expected that due to the decreased complex stability, the K_d would also be weakened by TFIIIF. In our experiments, we measured TFIIIF-induced dissociation rate changes directly but not K_d s, because it was previously shown that TFIIIF increases dissociation of non-repressive ncRNAs (Alu-LA, B1 RNA) but not repressive ncRNAs (B2, B1-Alu chimeric RNAs), in every case tested. The reviewer is correct that we have not assessed K_d directly, or whether K_{on} may also be affected, especially in the context of TFIIIF mutants.

Because of challenges in producing sufficient amounts of TFIIIF mutant proteins, and because Pol II-TFIIIF complexes were not stable at lower concentrations, additional K_d measurements were not feasible. During the revision period, we discovered that the Pol II-TFIIIF dissociation assays were extremely sensitive to buffer conditions, so we have now removed our data for TFIIIF-induced dissociation. We now focus on the function of the TFIIIF RAP30 and RAP74 C-terminal domains as revealed through the transcription assays, and have added new experiments testing of the ability of TFIIIF and various TFIIIF mutants to bind to Alu RNA. These new data show that TFIIIF is able to directly bind Alu-LA and Alu-RA at nM concentrations, suggesting that TFIIIF-RNA binding may play a role in relief of Alu RNA repression.

Ideally, I would prefer using non-structured RNA of similar length as a negative control(in addition to the "no RNA") as there could be non-specific effect of adding polynucleotides.

Thank you for the suggestion. We have performed an additional transcription experiment in the presence of a 151-nt long RNA, which was predicted by RNAfold to be largely unstructured. These experiments revealed a modest repression similar to that observed with Alu-LA in the presence of TFIIIF and are now included in Extended Data Fig. 6d. This result is in line with previous observations that unstructured RNAs nonspecifically inhibit Pol II transcription (Pai et al. RNA. 2014), and with the activity of Alu-RA as a strong transcriptional repressor. We note that the lanes supplemented with RNA also include an additional, DNA template-independent product, which could not be completely removed by a second step of DNase I treatment and RNA repurification.

>"We observed similarly high affinities for Alu-LA and Alu-RA when investigated in the presence of 150 mM monovalent salt, with the non-repressive Alu-LA binding with slightly higher affinity (Fig. 3a)."

Does this mean that these assays were performed with 150mM salts?

In the methods section, 50 mM KCl (or NaCl) buffers are listed.

Please clarify.

Thank you for pointing this out. This buffer description had been carried over from an earlier version of the manuscript, and we have now corrected it. The binding and dissociation assays were carried out in the presence of 150 mM NaCl. We have corrected this in the methods description. In addition, we have updated the methods text to reflect experiments repeated during the revision using a more efficient method of RNA fluorescent labeling.

>Cryo-EM sample preparation

The authors seem to use zero Mg²⁺ buffer for the structural analysis.

This reviewer is not completely sure if this is ok-ish, because Mg²⁺ can be important for RNA structures and/or RNA-protein interactions, and the authors include 4mM Mg²⁺ for assays.

From initial RNA folding tests, we were not able to detect any effect of Mg²⁺ on the folding of Alu RNA segments as detected by electrophoretic mobility shift assay. Indeed, the majority of Alu-RA and all of Alu-LA migrated as a more compact, folded species as purified after in vitro transcription, without any additional RNA folding procedure.

To address this potential concern, we have performed RNase T1 footprinting on Alu-LA and Alu-RA that have been (re-)folded in various concentrations of Mg²⁺, and we were not able to detect any difference in the cleavage pattern, suggesting that the base pairing of Alu RNA is unchanged under these varying conditions (Response Figure 2a). Additionally, we have repeated the Pol II-Alu RNA affinity measurements in the absence of Mg²⁺ and show that the affinity between Pol II and Alu RNA is not negatively affected by the lack of additional magnesium (Response Figure 2b). Therefore, if there are any Mg²⁺-dependent differences in folding, they are unlikely to affect regions of Alu RNA in contact with Pol II.

Response Figure 2. The effects of magnesium ion concentration on Alu RNA folding and Pol II binding. (a) RNase T1 footprinting of Alu-LA (left) and Alu-RA (right). Denaturing urea-PAGE analysis (sequencing gel) of RNA products, visualized using an Amersham Typhoon RGB to detect the 3' ATTO 488 label. Lane 1, no RNase T1 control; lane 2, no folding; lanes 3-4, no Mg²⁺, lanes 5-6, Mg²⁺ as indicated. Folding temperature indicates the maximum temperature to which the RNA was heated before cooling. Folding speed, F indicates fast folding (placement on ice directly after heating), S indicates slow folding (controlled cooling of 1°C/30s to 4°C). Different folding procedures all yield similar RNase T1 digestion patterns. (b) Fluorescence anisotropy binding experiments using ATTO 488-labeled Alu-LA and Alu-RA. Experiments were carried out in the buffer conditions for cryo-EM sample preparation, either with or without the addition of 4 mM Mg²⁺. Buffer solution base: 5 mM HEPES-KOH pH 7.25, 150 mM NaCl, 10 μM ZnCl₂,

10 mM DTT. The interaction between Pol II and RNA is not negatively affected by the omission of Mg^{2+} from the buffer. Under these buffer conditions, magnesium ions appear to slightly weaken the Pol II-RNA interaction.

Also, the reviewer wonders if the active site Mg^{2+} depicted in Fig2d was observed or not.

We did indeed observe density for an active site Mg^{2+} in the Pol II-minimal Alu-RA RNA map (Response Figure 3).

Response Figure 3. Density for an active site Mg^{2+} ion in the Pol II-minimal Alu-RA alternative fold map. Zoomed-in region of the minimal Alu-RA alternative conformation model overlaid with the EM density (DeepEMhancer post-processed for visualization). The three active site aspartates are labeled in light gray (D495, D497, and D499), the 3' end of the minimal Alu-RA model is shown in dark gray, and the magnesium is shown in magenta.

>Cryo-EM data processing

The authors performed masked classification with RNA(/clamp) masks for the Alu-LA and minimal Alu-RA datasets.

However, no such step seemingly exists for the Alu-RA dataset.

This reviewer wonders why this is the case, because consistent data processing is usually preferred if they want to compare structures from different datasets.

We did perform masked classifications using RNA and RNA/clamp masks for the Alu-RA data set, but did not include this in Extended Data Fig. 3, as neither the density for the RNA nor the nominal resolution of the reconstruction were significantly improved. In the case of the Pol II-Alu-LA map, the nominal resolution of the map (and the Pol II density) was slightly improved, whereas the RNA density remained very similar.

For comparison, in Response Figure 4 below, we show the densities obtained prior to masked classification (a) and after classification using an RNA/clamp mask (b) for both Pol II-Alu-LA and Pol II-Alu-RA, filtered to 8 Å resolution as displayed in the figures. The maps pre- and post-RNA/clamp classification are very similar.

Response Figure 4. Pol II-Alu RNA densities before and after classification using an RNA/clamp mask are very similar. (a) Pol II-Alu-LA and Pol II-Alu-RA densities prior to masked classification. Alu-LA, orange; Alu-RA, purple. Particle numbers are shown underneath each density. (b) Pol II-Alu-LA and Pol II-Alu-RA densities after classification using an RNA/clamp mask. Coloring and labeling as in (a).

This reviewer also wonders if running more 3D classifications could help improve the RNA density of the Alu-RA dataset, as there are more than 800k particles in the final reconstruction.

Thank you for this comment. This was also our original hope. We performed extensive 3D classification in Relion (various masks with various hard and soft edge pixel counts, different T values (2-20), different resolution limits (no limit up to 18 Å)). We additionally performed 3D classification in Cryosparc using up to 100 classes, as well as Cryosparc 3D variability analysis using several different masks. Finally, we used cryoDRGN to try to sort out a subset of data with improved resolution for the RNA. Unfortunately, none of these analyses led to an improved RNA density, although they did suggest the presence of extremely variable density near the upstream DNA-binding region and RNA exit. We did not include any of these classifications in the Alu-RA processing tree, as there were no detectable improvements in the interpretability of the RNA. We therefore suspect that our analysis is limited by the intrinsic flexibility of the RNA.

Minor comments:

>We solved the structure of the Pol II-minimal Alu-RA complex to a nominal resolution of 3.1 Å, and observed that the quality of the RNA density was greatly improved (Fig. 2a)

This statement could be a bit misleading, as the "alternative conformation" is shown in fig2a before it is described in the next sentence. Also because it looks like the quality improvement is only in the alternative conformation, and not in the canonical conformation.

In the Pol II-minimal Alu-RA complex structure, we did observe improved density prior to the 3D classification and modeling that revealed that the well-resolved RNA was in an alternative conformation, likely because the lower resolution canonical conformation RNA density was averaged out. We have rephrased the sentence and moved the reference to Fig. 2a to the following sentence to decrease the chance for misunderstanding.

“We solved the structure of the Pol II-minimal Alu-RA complex to a nominal resolution of 3.1 Å, and observed that the quality of the RNA density **in the initial map** was greatly improved (Fig. 2a).”

> Altogether, despite Alu RNAs adopting an EC-like conformation, the Pol II enzyme was observed in an inactive state, consistent with the repressive activity of Alu RNA.

This reviewer doubts if the fork-loop conformation is **consistent with** the repressive activity of Alu RNA.

(This reviewer assumes that if Alu RNA binds the cleft, the polymerase can not bind the promoter or "correct" DNA, and this is why transcription is repressed. If so, the loop conformation does not seem to be relevant for the repression, unless PolII is extending the Alu RNA.)

We agree and have rephrased this statement. The inactive state we observed is, however, consistent with a non-transcribing Pol II enzyme, suggesting that, unlike the bacterial 6S RNA or mouse B2 RNA, Alu RNA would not be a template for Pol II elongation.

We have changed the phrasing as follows:

“Altogether, despite Alu RNAs adopting an EC-like conformation, the Pol II enzyme was observed in an inactive state, consistent with ~~the repressive activity of Alu RNA~~ an enzyme-nucleic acid complex not competent for elongation.”

>Purified RNA was refolded by heating to 95°C for 5 min in buffer as specified

Please specify the buffer composition.

(Possibly the authors used 150mM NaCl buffer for cryo-em, and 50mM KCl buffer for assays?, but please make it clearer.)

We have clarified this in the methods, and apologize for the confusion. We have used 150 mM NaCl in the buffer for the assays of the original submitted manuscript.

>S. scrofa Pol II was purified as previously described

Please specify the composition of the Pol II storage buffer.

Note that the reviewer is interested if they have residual Mg²⁺ somewhere.

The Pol II storage buffer contains: 5 mM HEPES-KOH pH 7.25 (25 °C), 150 mM NaCl, 10 μM ZnCl₂, 10 mM DTT. The Pol II active site Mg²⁺ remains bound throughout the purification. This same buffer has been used previously in the determination of cryo-EM structures of Pol II elongation complexes, in each of which the active site Mg²⁺ could be resolved (Bernecky et al. Nature. 2016; Bernecky et al. NSMB. 2017). It is unknown whether additional Mg²⁺ ions co-purify with the endogenous Pol II complex.

>HEPES (many times)

HEPES-NaOH or HEPES-KOH ?

We have clarified this in the methods.

>Pol II was preincubated with TFIIF or TFIIF mutants (protein:RNA molar ratio of 1:4) possibly, PolII:TFIIF molar ratio

We thank the reviewer for catching this. We have corrected this in the methods text.

>, then Alu RNA was added RNA to the reaction
typo?

Yes, we have corrected this in the methods text.

>Cryo-EM stats

Please add phase randomized FSCs to extended figures.

Please add number of micrographs and dose rate(e-/s/px) to Table1.

We have added these items to Extended Data Figures 2, 3, and 4 and Table 1.

Reviewer #2:

Remarks to the Author:

The brief communication by Tlučková et al investigates the molecular mechanism of transcriptional inhibition by Alu RNA. The study is an extension of the original work by Goodrich and coworkers who identified this RNA as contributing to the inhibition of transcription initiation during heat shock and characterized the importance of smaller RNA segments and segment features in the repression mechanism using in vitro transcriptional reconstitution studies. Tlučková et al now aim to place that work and further biochemical characterization into a structural framework of PolII-Alu-RNA interactions and how they compete with normal nucleic acid engagement in transcribing Pol II complexes. The work includes both cryo-EM studies with a number of different ALU RNA constructs, inspired by Goodrich's work, as well as biochemical and transcriptional assays. The structural work extends that published 10 years ago at much lower resolution, which already concluded the RNA was occupying the cleft of Pol II where the template DNA and the RNA-DNA hybrid should be located during initiation and elongation of Pol II.

I find the present work informative and leading to a refinement of the original models, including more quantitative data and higher resolution structures that now can try to place the biochemical findings into an structural framework that better explains them. I would support publication if the authors were to consider certain changes in their discussion and in the presentation of their data that could make the data presented and the models proposed more compelling.

We thank the reviewer for their assessment and support, which helped us improve our manuscript.

The resolution on the Alu-RNA for the larger constructs is much lower than for Pol II itself. This could be because under the conditions used the occupancy of the RNA is low, or because there is heterogeneity in the RNA mode of binding (e.g. multiple registers with respect to the protein), or

due to the flexibility of the RNA itself. The image analysis pipeline describes the process of 3D classification using a mask for the clamp and RNA region, but it is not clear what the findings were. For example, what were the difference between the two major classes, 2 and 7? Were the other “empty” classes with no RNA? In any case, even after this sorting, variability still reduce the resolution of the RNA, indicating either that the contacts with the protein are not extensive or that there are multiple modes of interaction. Could this be actually important for Alu-RNA inhibition?

Given the high affinity of the Pol II-Alu complexes, and the fact that we did not observe any high-resolution 3D classes completely lacking nucleic acid/Pol II cleft density, we expect that the occupancy of the RNA is high. For each of the data sets, the first global classification revealed reconstructions of lower or higher quality, which may reflect variations in ice thickness and/or particle quality in different regions of graphene oxide coverage. 3D classifications using a clamp mask revealed that for the Alu-LA and Alu-RA data sets, the clamp could be observed in multiple conformations, but these conformations did not appear to be correlated with quality of RNA density. 3D classifications using an RNA-region mask revealed the largest variability in densities near the upstream DNA-binding region and RNA exit.

Our hypothesis is that the lower resolution of the RNA is due to the flexibility of the RNA, both in its tertiary structure, but also in its base-pairing propensity in the regions of imperfect complementarity. These unpaired bases would be expected to interact weakly with neighboring bases, and in a promiscuous manner. Thus, there could be many very similar Alu RNA molecules which could each interact slightly differently with Pol II.

We would suggest that for RNA inhibition, it may be important for an RNA to either have flexibility to adopt an EC-like structure, or alternatively for it to adopt a rigid structure that closely mimics EC nucleic acids. Furthermore, we would postulate that the “imperfect” mimicry of Alu RNA may be advantageous, in that the repression of Alu RNA can be overcome (RNA can be displaced) without the requirement for RNA elongation (and the resulting alteration of the RNA structure), as is utilized in the bacterial 6S RNA system.

Can the authors comment a bit more on the differences (or not) of the structures for the two Alu segments, given that they do have different activities?

We did not observe any differences for the two Alu segments that we are confident to include in the manuscript, given the lower local resolution of the RNA. While the Alu-RA density does appear to extend slightly longer than the Alu-LA density in the direction of the downstream DNA, given the low resolution of this region and the larger data set size for Alu-RA, we decided not to comment on this in the manuscript.

The authors refer briefly to the difference in conformation of the Fork2 loop in their structure of PolII-Alu-RA versus that in the elongating complex. Unfortunately, the corresponding figure is tiny and buried in a supplemental figure. Can they show whether the change in the Alu-containing structure is not due to contact with the Alu RNA, rather than simply due to lack of contact with a non-template strand? In that paragraph the authors state “Altogether, despite Alu RNAs adopting an EC-like conformation, the Pol II enzyme was observed in an inactive state,

consistent with the repressive activity of Alu RNA.” That conformation seem irrelevant to the repression process compared with the occupancy of the critical binding sites for the DNA and DNA-RNAs hybrid by the Alu-RNA!! Are the authors implying that the Alu RNA captures an inhibited state? That somehow that state will be relevant for competing off the inhibitory RNA? I fail to see the significance.

We apologize for the confusion. Because fork loop 2 is also rearranged in a similar way in the structure of Pol II lacking any nucleic acids, we believe that the lack of nucleic acid contact is the most likely explanation for this rearrangement. This observation suggests that unlike the bacterial 6S RNA, Alu RNA does not seem to be competent for elongation by Pol II. We have changed the text to say:

“Altogether, despite Alu RNAs adopting an EC-like conformation, the Pol II enzyme was observed in an inactive state, consistent with ~~the repressive activity of Alu RNA~~ an enzyme-nucleic acid complex not competent for elongation.”

When working with the minimal domain, it would be informative to show more clearly how different, other than in resolution, are RNA densities in the structure for the “canonical” and “alternative” conformation. Could they be shown side by side, or even superimposed, maybe by themselves and with the model of the nucleic acid in the EC? (for example, do the equivalent of Sup Fig 4g for the alternative, right next to that panel).

We have included this figure as suggested (revised Extended Data Fig. 4g).

Can the authors speculate how the RNA could “switch” between one state and the other?

Rather than switching, we believe it is more likely that the RNA folds into distinct populations during the folding procedure. We speculate that the alternative conformation of minimal Alu-RA may be favored in the truncated construct. Notably, this unexpected finding did strengthen a key insight of our study, namely that mammalian Pol II efficiently binds RNA that is able to mimic EC nucleic acids rather than promoter DNA nucleic acids.

It seems that the canonical minimal state shows less density than the full Alu-RA. Can the authors use that to speculate which regions of the RNA density correspond to what part of the molecule?

Thank you for the suggestion. In the revised manuscript, we now propose a rough assignment of the regions of Alu-RA that bind in the DNA-RNA hybrid and the downstream DNA-binding regions of the Pol II cleft (Extended Data Fig. 4i). We note that these are putative assignments owing to the limited resolution of the RNA densities.

Also, are the authors sure that the alternative conformation was not present for the RA or RL structures? Did they look for it among the discarded particles in those data sets?

Despite extensive 3D analysis of our cryo-EM data sets, particularly the Pol II-Alu-RA data set, we did not observe anything resembling the alternative conformation in the Alu-RA or Alu-LA

data. Given the large size of the Pol II-Alu-RA data set and the observation that the alternative conformation was clearly resolved in the Pol II-minimal Alu-RA reconstruction even before classification, we believe it is unlikely to be present in our Alu-LA or Alu-RA data sets at a detectable level.

I am not sure of the intent and value of the speculation in lines 105 to 115, irrespective of the mimicry of one of the conformers of a minimal Alu-RNA to the DNA-RNA hybrid. The original work of Goodrich established that Alu can only inhibit if the PolII-Alu-RNA complex is formed previous to Pol II recruitment to the promoter (presumably the PIC), at which point the loading of Pol II onto the promoter takes place and repression does not happen.

We do not mean to comment on the timing of Alu RNA repression within a cell, but rather we mean to speculate on the characteristics of any non-coding RNA that would be active in repression of mammalian Pol II. In the paper from the Goodrich lab (Mariner et al. Mol Cell. 2018), the authors speculate that the “loose” region within Alu RNA allows Alu to mimic an open promoter DNA complex, as was observed for bacterial 6S RNA. With this section, we wish to rationalize why, as we observe for Alu-LA, Alu-RA, and both conformations of minimal Alu-RA, the mimicry of open promoter DNA is not conserved from bacteria to humans. Importantly, our work suggests that mimicry of a Pol II elongation complex rather than open promoter DNA complex is the mechanism by which Pol II can be efficiently repressed.

We have rephrased the text to clarify (edits to the referenced paragraph in red).

“Analogous to Alu RNA, bacterial 6S RNA is a natural noncoding RNA that directly binds the bacterial RNA polymerase and inhibits transcription from $\sigma 70$ -dependent housekeeping promoters. **Both Alu and 6S RNA inhibit promoter DNA engagement.** A previous near-atomic structure of *E. coli* RNA polymerase in complex with 6S RNA showed that this RNA adopted a conformation mimicking that of open promoter DNA¹⁰. This contrasts with our observation that Alu RNA mimics elongation complex nucleic acids (Fig. 1c). Previous results have shown the importance of a loosely structured region within Alu-RA for repressive activity² (Fig. 1a). We conclude that **loosely base paired regions within Alu RNAs are** important not as a mimic of open promoter DNA (Extended Data Fig. 5c), but **speculate that it** allows the RNA **the flexibility to** adopt a conformation matching that of the elongation complex DNA-RNA hybrid and downstream DNA. This is consistent with previous observations that, unlike bacterial promoter complexes which can form a stable complex with just RNA polymerase and a sigma factor^{11,12}, eukaryotic initiation complexes require a large number of transcription factors to stabilize the interaction with DNA¹³ and are dynamic¹⁴. Promoter complexes lacking accessory factors or a nascent RNA chain are unstable¹⁵. Thus, the more stable elongation complex¹⁵⁻¹⁷ would be an effective mammalian intermediate to mimic.”

The authors try to address important functional questions, such as the fact that Alu-LA is more labile than Alu-RA in the presence of TFIIF, in spite of both forming similar structures, via biochemical assays. They found that TFIIF markedly reduced the half-life of Alu-LA on Pol II, with little effect on that of Alu-RA. The authors then postulate that Alu-LA is less stable in the

bent conformation required to bind in an EC-like conformation, and therefore more labile and more susceptible to dissociation from Pol II bound to TFIIF. That implies a shorter half life even in the absence of TFIIF than that of Alu-RA, which they showed is not the case. Whatever the mechanism, it has to be one that manifest itself only in the capacity of TFIIF to actively remove one construct versus the other. Thus, the authors need to come out with a better hypothesis of why the additional flexible element in Alu-RA makes it less susceptible to be competed off by TFIIF.

We have now edited the manuscript to focus on the effects of TFIIF domains on Alu-LA and Alu-RA activity. We have replaced the dissociation measurements with TFIIF-RNA binding assays, because of (1) experimental constraints regarding the analysis of TFIIF mutants, (2) experiments completed during the revision period that have shown that Pol II-TFIIF dissociation rates were very sensitive to buffer conditions, and (3) to keep within the length constraints of the NSMB Brief Communication. In brief, the newly added TFIIF-RNA binding experiments show that TFIIF is able to directly bind Alu-LA and Alu-RA at nM concentrations, suggesting that TFIIF-RNA binding may play a role in relief of Alu RNA repression. In place of the Alu-RA “loose” region, Alu-LA contains a smaller predicted mismatched region followed by a bulge, which could result in less effective downstream DNA mimicry. This would be consistent with our observation that Alu-RA density may extend slightly longer than the Alu-LA density in the direction of the downstream DNA (see Response Figure 4).

We hope that the newly presented discussion is satisfactory for the reviewer.

The authors indicate that TFIIF would not sterically clash with the resolved RNA elements. Unless their density can account for 100% of the nucleotides present in the MINIMAL construct, this argument cannot exclude steric hindrance with elements that are not visible, and in particular, with elements that are different between Alu-LA and -RA.

Thank you for this comment. We do observe that the Pol II-alternative Alu-RA cryo-EM density can account for the entire RNA construct, whereas the Pol II-canonical minimal Alu-RA cryo-EM density could account for the majority (~80% depending on visualization threshold) of the minimal Alu-RA RNA sequence (Response Figures 1 and 5). The remaining canonical conformation nucleotides would project towards the downstream DNA binding region, where no regions of TFIIF have been observed in human or yeast Pol II initiation complex structures from the Cramer, Nogales and Xu labs. We therefore do not expect that steric exclusion due to Pol II-TFIIF contacts is a major driver of the ability of TFIIF to relieve Alu-LA repression.

Response Figure 5. Speculative, complete model of the minimal Alu-RA alternative conformation and the observed density. The model of 10-subunit Pol II is shown in the left panel for orientation. The alternative conformation difference density, filtered to 5 Å (dust hidden) is shown in transparent dark gray. The alternative conformation model of minimal Alu-RA was generated by appending regions 7-47 and 63-68 of farfar2-generated models (Lyskov et al PLoS One. 2013) onto the minAluRA alternative conformation structure shown in Figure 2.

The authors carry out transcription repression assays in the presence of TFIIF, using both wild type and mutants. I find the explanation for the results using the TFIIF (RAP74DC) mutant highly confusing. An important question to be addressed was why, in spite of their similar binding, only the Alu-RA half can inhibit transcription in the presence of TFIIF? Have their data really explained this behavior?

In response to these comments and those of Reviewer 1, we have substantially revised the section dealing with biochemical assays. We have removed the Pol II-RNA dissociation assays, and instead focus on the transcription assay results. We additionally demonstrate the direct interaction of TFIIF with Alu-LA and Alu-RA. From these results, we propose a mechanism by which TFIIF may more strongly affect Alu-LA. Specifically, in Extended Data Fig. 6f and g, we show that TFIIF binds directly to Alu-LA and Alu-RA and that interactions between Alu RNAs and TFIIF are reduced for the TFIIF (RAP74ΔC) mutant, particularly for Alu-LA. We now propose that direct TFIIF-RNA binding may be important for relieving Alu RNA repression. We hope the referee agrees with these changes.

Transcriptional repression involves Alu RNA incorporation with Pol II into stable complexes at promoters while Alu RNA does not repress transcription after Pol II has formed preinitiation complexes. These observations led Goodrich and coauthors to suggest that once Pol II is engaged with promoter DNA it is resistant to repression by Alu RNA. This can now be explained by the present work and the authors should emphasize it.

We appreciate the reviewer's assessment and have edited the text to include the following statement:

"The observed density localization explains how Alu RNA would block productive engagement with promoter DNA during transcription initiation, but could still allow assembly of a preinitiation complex of altered topology through TFIIB- and TBP-promoter DNA contacts."

On the other hand, something that the authors do not mention, perhaps because their work cannot address it, is that two distinct regions of Alu RNA that function to mediate transcriptional repression are not required for Pol II binding (again in reference 2). In other words, the binding is not sufficient for repression. These regions, maybe not visible in EM structures, may be

involved in precluding Pol II engagement with the PIC, thus avoiding competition of promoter DNA for Alu RNA. Maybe the authors have a better speculation to make in the context of their work?

Thank you for the comment. The work by Mariner et al. (reference 2) showed that two Alu regions are important for repression: the loose region of Alu-RA, and the A-rich linker between Alu-LA and Alu-RA.

In our work, we have resolved the loose region of Alu-RA, as it is contained within the minimal Alu-RA construct and is most likely part of the RNA region binding near the active site and downstream DNA-binding regions of Pol II. The A-rich linker between Alu-LA and Alu-RA was not included in the constructs used for cryo-EM, but we would speculate that it may be involved in non-specific interactions that increase the affinity of ncRNA for Pol II. If our putative assignment shown in Extended Data Fig. 4i is correct, the A-rich linker would be placed in close proximity to the RNA exit channel, and if bound there, could contribute to the affinity and/or stability of the Pol II-RNA complexes. Because this region is not able to bind to Pol II on its own (Mariner et al. Mol Cell 2008.), it is unlikely to drive Pol II-ncRNA association. Because this is quite speculative, we would prefer not to comment on the putative role of the A-rich repressive region, but can add it if the reviewer desires.

List of responses to reviewer comments

“Mechanism of mammalian transcriptional repression by noncoding RNA”
(Katarína Tlučková, Beata Kaczmarek, Anita Salmazo, and Carrie Bernecky)

NSMB Manuscript NSMB-BC48182

Responses are in *italics*.

Reviewer comments:

Reviewer #1:

Remarks to the Author:

In this revised manuscript, the authors have made substantial changes to the text. The manuscript is generally improved, with less speculations, better readability and better presentations of data. Also, the authors included a new binding assay of TFIIF and Alu RNA, trying to conclude that the higher binding affinity of TFIIF RAP74dC to Alu-LA (compared to the affinity to Alu-RA), is related to the functional difference between Alu-LA/RA .

Comments:

>Specifically, in Extended Data Fig. 6f and g, we show that TFIIF binds directly to Alu-LA and Alu-RA and that interactions between Alu RNAs and TFIIF are reduced for the TFIIF (RAP74DC) mutant, particularly for Alu-LA. We now propose that direct TFIIF-RNA binding may be important for relieving Alu RNA repression.

This reviewer is rather cautious about this statement, because

1. It is hard to judge from the graphs(ExFig6g) if the difference between LA/RA is statistically significant.
2. Even if it is significant, the difference does not look very large (about 2-fold affinity difference?)
3. At 60nM concentrations, there are multiple bands of complexes, which suggests that multiple TFIIF molecules can bind to the RNA(ExFig6f). Also, the amount of upper band looks very different between TFIIF variants, or between Alu-LA/RA. As transcription assays are performed with much higher concentrations (120nM Pol2-480nM TFIIF-480nM RNA?), the actual situation could be much more complicated.

This reviewer is satisfied with other responses.

We thank the reviewer for their additional comments. We fully agree that the situation in the context of a Pol II-TFIIF-Alu RNA complex, a full transcription initiation complex, or indeed within cells may be more complicated.

According to a two-tailed unpaired t-test, the differences are significant in both cases. In the case of Alu-RA, the difference between binding to wild-type TFIIF and TFIIF (RAP74 Δ C) does not reach the level of significance for the 15 nM measurement, but is significant for the 30 nM measurement ($p=0.0020$). In the case of Alu-LA, the difference between binding to wild-type TFIIF and TFIIF (RAP74 Δ C) is significant at both concentrations (15 nM, $p= 0.0001$; 30 nM, $p= 0.0017$). We wished to comment on this slight difference in the text, but now see that this statement may be misleading. We have deleted the phrase “particularly for Alu-LA” from the text.

Our main conclusion from the RAP74 Δ C and Alu RNA binding assays is that deletion of the RAP74 C-terminal region decreases RNA affinity, and therefore this region contributes to TFIIF binding to Alu RNA. This observation suggests that it is possible that the RAP74 C-terminal region may bind to RNA and affect its association with Pol II in the context of a Pol II-Alu RNA complex, depending on the length and structural features of the 3' region of the RNA.

Reviewer #2:

Remarks to the Author:

I find that the authors have made a good attempt at answering the reviewers comments and that the paper is improved enough to warrant publication

We thank the reviewer very much for their help during this revision process.